# Temporal analysis of two inducible human genes reveals post-transcriptional H3K4me3 deposition

Komal Paresh Walvekar[1,2], Sabarinadh Chilaka[1,2,3]

**Histone H3 lysine 4 trimethylation (H3K4me3) is an established hallmark of active promoters, yet the temporal hierarchy between its deposition and transcriptional initiation remains incompletely understood. Here, we employ temporal dynamic analysis of inducible human inflammatory genes to demonstrate that H3K4me3 accumulation lags significantly behind transcriptional onset. At the *TNF-α* and *IL-1β* loci, H3K4me3 enrichment succeeds the rapid recruitment of RNA polymerase II, NF-κB, and p300, appearing kinetically decoupled from the initial establishment of histone acetylation. We find that H3K4me3 deposition is attenuated by transcriptional inhibition, characterizing it as a downstream event rather than a prerequisite for initial activation. Furthermore, MLL1-mediated reduction of H3K4me3 does not impair transcriptional induction, demonstrating that this mark is dispensable for the primary onset of expression at these loci. At the constitutively active *MYC* locus, H3K4me3 persists after transcriptional inhibition, indicating temporal uncoupling between transcription and H3K4me3 maintenance. Our findings provide a detailed kinetic characterization of H3K4me3 loading and suggest that at these inducible genes, this modification serves as a consequence of, rather than a trigger for, transcriptional initiation.**

## Introduction

Histone post-translational modifications (PTMs) regulate chromatin structure and gene expression, governing processes ranging from embryonic development and cell lineage specification to genome stability, DNA repair, and the onset of pathological conditions (Zhou et al, 2011; Allis & Jenuwein, 2016; Talbert & Henikoff, 2021; Millán-Zambrano et al, 2022; Weinzapfel & Fedder-Semmes, 2024). H3K4me3 is among the extensively studied histone modifications, with more than 4,000 publications in PubMed highlighting its importance in epigenetics. H3K4me3 is ubiquitously found at the promoters of actively transcribed eukaryotic genes, closely associated with open chromatin regions, and is often dysregulated in several diseases, including cancer (Santos-Rosa et al, 2002; Schneider et al, 2004; Bernstein et al, 2005; Pokholok et al, 2005). Although H3K4me3 has been considered as a hallmark of active transcription, its precise role in transcriptional regulation remains poorly understood.

Given its strong association with actively transcribed promoters, H3K4me3 is often implicated in facilitating transcriptional initiation, but its direct functions are not yet fully understood (Vermeulen et al, 2007; Lauberth et al, 2013). On the other hand, increasing evidence from diverse contexts shows that transcription can occur without detectable H3K4me3, suggesting it is not always essential for transcription initiation (Carlone et al, 2005; Clouaire et al, 2012; Ding et al, 2012; Hödl & Basler, 2012; Soares, 2012; Wang et al, 2023). These reports highlight a striking paradox: although H3K4me3 is strongly associated with active promoters, it appears dispensable for transcription initiation. These observations raise an important question: does H3K4me3 actively drive transcription, or is it simply a consequence of transcriptional activity merely acting as a mark of active transcription (Henikoff & Shilatifard, 2011; Howe et al, 2017; Talbert & Henikoff, 2021; Yu & Lesch, 2024; Wang & Helin, 2025). Addressing this uncertainty regarding such a prevalent and well-studied histone mark has broad implications, from advancing our understanding of fundamental gene activation mechanisms to informing therapeutic strategies, particularly in human diseases such as cancer, where H3K4me3 is frequently dysregulated.

In this study, to address the H3K4me3 paradox, we employed time-resolved analyses of transcriptional activity and histone modifications at inducible human *tumour necrosis factor-alpha* (*TNF-α*) and *interleukin-1 beta* (*IL-1β*) loci. We demonstrate that H3K4me3 accumulation occurs with a distinct kinetic delay relative to the onset of transcription, lags behind the recruitment of key factors, and follows the establishment of promoter acetylation. Importantly, H3K4me3 recruitment depends on prior transcriptional activity, whereas transcription itself proceeds independently of this modification. Furthermore, transcription blocking assays at the constitutively expressed *MYC* (*MYC* proto-oncogene) gene reveal a similar temporal uncoupling, indicating that this kinetic disparity may extend beyond inducible inflammatory

[1]Division of Applied Biology, CSIR-Indian Institute of Chemical Technology, Hyderabad, India   [2]Academy of Scientific and Innovative Research (AcSIR), Uttar Pradesh, India   [3]DBT-Ramalingaswami Re-entry Fellow, Department of Biotechnology (DBT), Regional Centre for Biotechnology, Faridabad, India

Correspondence: sabarinadhch@gmail.com, sabarinadh@csiriict.in

models. Together, our results demonstrate that H3K4me3 accumulation follows rather than precedes transcription, suggesting that in these distinct human gene models, the mark serves as a readout of transcriptional activity rather than an inductive trigger.

# Results

### Inducible human transcription models to investigate H3K4 methylation dynamics

To better understand the dynamic regulation of H3K4me3 during transcription initiation, we employed an inducible gene system as a precise and innovative model to study transcriptional activation. We propose that, unlike constitutively active gene models, inducible gene models provide precise control over gene off-to-on transitions, making them ideal for studying the dynamic regulation of histone post-translational modifications. Here, we employed the well-characterized human inducible inflammatory genes tumour necrosis factor-alpha (*TNF-α*) and interleukin-1 beta (*IL-1β*) to investigate histone dynamics associated with transcriptional activation. These genes undergo robust transcriptional induction in immune cells, such as monocytes and macrophages, upon stimulation with inflammatory agents like bacterial lipopolysaccharide (LPS) (Suzuki et al, 2000; Sharif et al, 2007; Adamik et al, 2013; Pulugulla et al, 2016). Human monocytic THP-1 cells were treated with LPS for various time points (0, 0.5, 2, 6, 12, and 24 h), after which they were harvested for parallel analysis of transcriptional activation and profiling of histone modifications at the corresponding promoters using quantitative reverse transcription PCR (qRT–PCR) and chromatin immunoprecipitation (ChIP) assays with real-time PCR (ChIP-qPCR), respectively (Fig 1A). We selected the promoter regions based on the high-resolution H3K4me3 ChIP-seq profiles from human monocytes (via the UCSC Genome Browser) and well-established genomic enrichment of H3K4me3 typically spanning −1 to +1 kb relative to the transcription start site (TSS) (Benayoun et al, 2014; Yu & Lesch, 2024). The genomic architecture and specific regions targeted for qRT–PCR and ChIP assays are detailed in Fig S1A–C. qRT–PCR analysis showed a significant up-regulation of *TNF-α* and *IL-1β* mRNA levels compared with 0 h, with peak expression observed at 2 h after LPS stimulation (Figs 1B and S2A). This transcriptional up-regulation was transient, as mRNA levels began to decline noticeably by 6 h and continued to decrease at 12 and 24 h. The observed rapid and transient expression patterns of *TNF-α* and *IL-1β* are consistent with their well-established temporal response to LPS stimulation (Sharif et al, 2007; Adamik et al, 2013; Pulugulla et al, 2016).

### H3K4me3 accumulation occurs with a distinct kinetic delay relative to transcriptional onset at human inflammatory genes

H3K4 methylation-comprising monomethylation (H3K4me1), dimethylation (H3K4me2), and trimethylation (H3K4me3)-is a well-characterized histone modification closely linked to active gene

transcription. H3K4me1 is commonly associated with active enhancers, H3K4me2 is predominantly enriched near the 5′ end of actively transcribed genes, and H3K4me3 typically marks the promoters of actively transcribed genes (Schneider et al, 2004). To elucidate the dynamic regulation of H3K4 methylation in relation to transcription initiation, we conducted chromatin enrichment analysis of H3K4me1, H3K4me2, and H3K4me3 at the *TNF-α* and *IL-1β* promoters during transcriptional activation (Fig 1). ChIP-qPCR analysis revealed a significant increase in H3K4me1 at the *TNF-α* promoter as early as 0.5 h after LPS induction, with elevated levels persisting at later time points (Fig 1C). Notably, H3K4me2 and H3K4me3 did not exhibit enrichment at the *TNF-α* promoter during the peak of transcription observed 2 h after induction (Fig 1D and E). Notably, strong and highly significant enrichment of both marks was observed at 6 h post-induction, despite a pronounced reduction in transcriptional activity (Fig 1D and E). To further support these findings, ChIP enrichment analyses were performed for additional histone marks associated with active transcription, including histone H3 lysine 27 acetylation (H3K27ac) and histone H4 lysine 8 acetylation (H4K8ac) (Wang et al, 2008; Stasevich et al, 2014). ChIP-qPCR results showed a significant increase in H3K27ac and H4K8ac marks at 2 h, coinciding with peak *TNF-α* mRNA expression (Fig 1F and G). This aligns with the well-established role of histone acetylation in promoter activation during transcription, and further validates our transcriptional model. To confirm the specificity of the observed signals, species-matched normal IgG was employed as a negative control, with all targeted enrichments showing minimal non-specific background (Fig 1H). Paralleling the chromatin dynamics of the *TNF-α* promoter, *IL-1β* activation was characterized by a coordinated sequence of histone modifications; specifically, acetylation synchronized with the transcriptional burst, whereas H3K4 di- and trimethylation lagged as post-transcriptional events (Fig S2B–G), further substantiating these findings. Furthermore, the dynamic H3K4 methylation profiles were corroborated at additional downstream regions, including exon 4 of *TNF-α* and intron 6 of *IL-1β*. These regions exhibited similar H3K4me1, H3K4me2, and H3K4me3 patterns, demonstrating a decoupled H3K4me3 peak at 6 h post-induction compared with the primary transcriptional peak observed at 2 h (Fig S3A–H). This finding reinforces the observation of post-transcriptional loading, where H3K4me3 accumulation follows transcription. Overall, these results demonstrate that H3K4me2 and H3K4me3 deposition did not correlate with the *TNF-α* and *IL-1β* transcriptional peak at 2 h, but showed a delayed enrichment post-transcriptionally at 6 h, irrespective of transcriptional activity.

### H3K4me3 displays temporal patterns distinct from other transcriptional markers at inducible genes

To further support the observation that H3K4me3 enrichment represents a post-transcriptional event uncoupled from transcriptional activity, we analysed whether the 6 h time point corresponds to a transcriptionally inactive state by examining the temporal dynamics of key transcriptional markers. Phosphorylation of the C-terminal domain (CTD) of RNA polymerase II (Pol II) at serine 5 (Ser5P) and serine 2 (Ser2P) serves as a well-established indicator of transcriptional activity (Hsin & Manley, 2012).

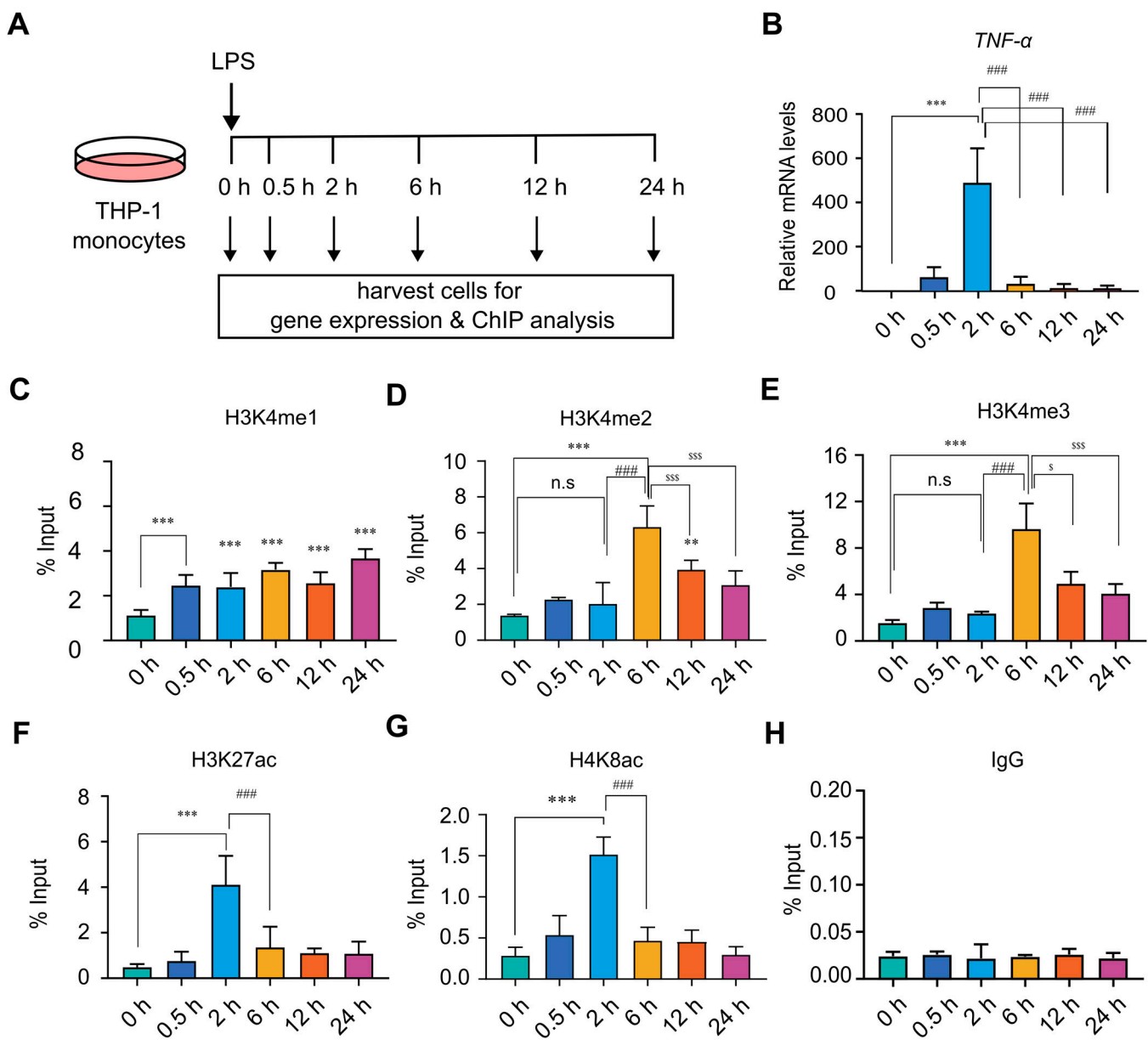

**Figure 1. Inducible gene activation model reveals temporal decoupling of H3K4me3 deposition and transcription.**
**(A)** Schematic representation of the experimental model used to study the temporal dynamics of histone post-translational modifications during LPS-induced transcriptional activation. Cells were stimulated with LPS (500 ng/ml) and harvested at 0, 0.5, 2, 6, 12, and 24 h post-stimulation for transcriptional and chromatin immunoprecipitation (ChIP) analyses. **(B)** qRT-PCR analysis of *TNF-α* mRNA levels after LPS stimulation, used as a readout of transcriptional activation. **(C, D, E, F, G, H)** ChIP enrichment analysis was performed using antibodies against (C) H3K4me1, (D) H3K4me2, (E) H3K4me3, (F) H3K27ac, (G) H4K8ac, and (H) IgG control. Immunoprecipitated DNA was analysed by qPCR using primers specific to the *TNF-α* promoter. Data are presented as % input, representing the enrichment of immunoprecipitated DNA relative to the total input chromatin. Statistical analysis was performed using one-way ANOVA followed by Dunnett's multiple comparisons test. Data are presented as the mean ± SD (n = 3) from independent biological replicates. Statistical significance is indicated as \*\*\*$P < 0.001$, \*\*$P < 0.01$, and \*$P < 0.05$ relative to the unstimulated control; ###$P < 0.001$, ##$P < 0.01$, and #$P < 0.05$ compared with LPS-stimulated cells at 2 h; \$\$\$$P < 0.001$, \$\$$P < 0.01$, and \$$P < 0.05$ compared with LPS-stimulated cells at 6 h; and ns as non-significant compared with the unstimulated control.

Specifically, Ser5P is associated with transcription initiation, whereas Ser2P marks the elongation phase. Chromatin enrichment analysis of Pol II Ser5P at the *TNF-α* and *IL-1β* promoters revealed a significant increase at the 2 h time point compared with the 0 h control (Figs 2A and S4A), consistent with active transcription initiation. However, Ser5P levels were significantly reduced at

6 h compared with 2 h, indicating a rapid down-regulation of transcriptional initiation activity (Figs 2A and S4A). Similarly, enrichment analysis of Pol II Ser2P at the *TNF-α* gene body (exon 4) showed a significant increase at 2 h, coinciding with the peak of transcriptional activity, followed by a significant decrease at 6 h compared with the 2 h (Figs 1B and 2B), suggesting a rapid

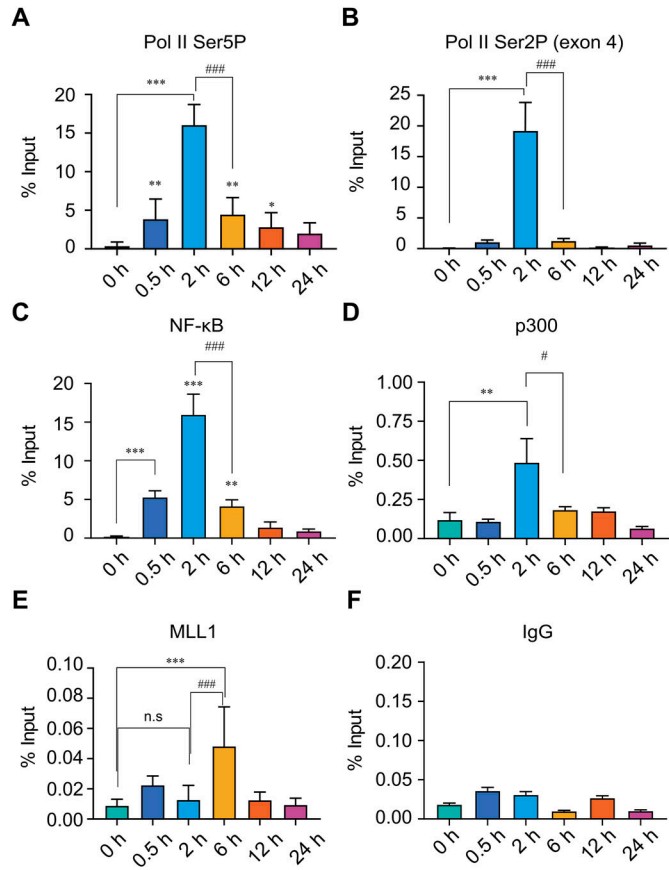

**Figure 2. ChIP enrichment analysis of transcriptional regulators at the *TNF-α* locus reveals temporal divergence from H3K4me3 during LPS-induced gene activation.**

THP-1 cells were stimulated with LPS (500 ng/ml) and harvested at 0, 0.5, 2, 6, 12, and 24 h to assess temporal occupancy of key transcriptional regulators at the *TNF-α* locus. **(A, B, C, D, E, F)** ChIP enrichment analysis was performed using antibodies against (A) RNA polymerase II Ser5-phosphorylated (Pol II Ser5P), marking transcriptional initiation; (B) RNA polymerase II Ser2-phosphorylated (Pol II Ser2P), marking elongation; (C) NF-κB (p65), a stimulus-responsive transcription factor; (D) MLL1, a histone methyltransferase responsible for H3K4me3 methylation; (E) p300, a histone acetyltransferase linked to transcriptional coactivation; and (F) IgG as a negative control. Enriched chromatin was quantified by qPCR using primers specific to the *TNF-α* promoter (A, C, D, E, F) or exon 4 (B). Statistical analysis was performed using one-way ANOVA followed by Dunnett's multiple comparisons test. Data are presented as the mean ± SD from n = 4 for (A, B) and n = 3 for (C, D, E, F) independent biological replicates. Statistical significance is indicated as ***$P < 0.001$, **$P < 0.01$, and *$P < 0.05$ compared with the unstimulated control; and ###$P < 0.001$, ##$P < 0.01$, and #$P < 0.05$ compared with LPS-stimulated cells at 2 h.

decline in transcriptional elongation. At later time points (12 and 24 h), both Ser5P and Ser2P levels remained low, indicating sustained transcriptional inactivity. To confirm assay specificity, we assessed Ser5P and Ser2P levels at the glyceraldehyde 3-phosphate dehydrogenase (*GAPDH*) promoter, a constitutively expressed gene commonly used as a positive control. As expected, both phosphorylation marks were consistently maintained at this locus (Fig S5A and B), validating assay specificity and supporting the reliability of the transcriptional dynamics observed at the target loci. Taken together, the comparison of RNA Pol II Ser5P and Ser2P enrichment dynamics with H3K4me3 supports the

conclusion that H3K4me3 loading at promoters occurs through a delayed post-transcriptional enrichment mechanism at these loci.

Inflammation-induced transcriptional activation of *TNF-α* and *IL-1β* is primarily mediated by nuclear factor kappa-light-chain-enhancer of activated B cells (NF-κB), a key regulator of inflammatory gene expression (Bhatt & Ghosh, 2014). In response to inflammatory stimuli, NF-κB translocates into the nucleus and activates transcription by binding to gene promoters and recruiting coactivators such as the histone acetyltransferase p300. To elucidate the temporal coordination between NF-κB–mediated transcriptional regulation and H3K4me3 deposition, we performed ChIP assays to assess NF-κB and p300 occupancy at the *TNF-α* and *IL-1β* promoters after LPS stimulation. NF-κB enrichment at the *TNF-α* promoter was detectable as early as 0.5 h post-stimulation and increased markedly by 2 h, coinciding with elevated transcriptional activity (Fig 2C). Notably, NF-κB occupancy declined significantly by 6 h compared with 2 h, correlating with reduced transcriptional activity, indicative of its transient promoter engagement. A similar pattern of NF-κB binding was observed at the *IL-1β* promoter (Fig S4B), reinforcing the rapid and dynamic nature of NF-κB recruitment correlating with transcriptional induction. Furthermore, p300 occupancy exhibited a similar recruitment pattern, with peak enrichment at 2 h post-stimulation (Figs 2D and S4C), aligning with both NF-κB binding and peak transcriptional activity. Together, these findings demonstrate that NF-κB and p300 are recruited during the early phase of *TNF-α* and *IL-1β* transcriptional activation, in contrast to the delayed deposition of H3K4me3. This temporal separation supports our finding that H3K4me3 functions as a downstream marker of active transcription rather than a prerequisite for its initiation and may be deposited through a mechanism independent of these transcription factors at these inducible gene loci.

### Mixed lineage leukaemia 1 (MLL1) recruitment correlates with post-transcriptional H3K4me3 enrichment

To independently confirm the timing of H3K4me3 deposition at the *TNF-α* and *IL-1β* promoters during transcriptional activation, we examined the recruitment of MLL1, a histone methyltransferase that catalyses H3K4 trimethylation at these loci. MLL1 was shown to promote inflammatory gene expression by H3K4me3 deposition downstream of NF-κB signalling, through its recruitment to NF-κB–bound promoters in monocytes and macrophages (Wang et al, 2012; Kimball et al, 2017; Wang et al, 2024). ChIP analysis showed no significant increase in MLL1 occupancy at the *TNF-α* and *IL-1β* promoters at 0.5 or 2 h after stimulation, as MLL1 occupancy levels remained comparable to those in unstimulated controls (Figs 2E and S4D). Notably, by 6 h post-induction, a significant increase in MLL1 occupancy was observed, which coincided with a pronounced enrichment of H3K4me3 (Figs 1E, 2E, and S4D). However, at later time points (12 and 24 h), MLL1 binding decreased, paralleling the decline in H3K4me3 levels. A normal rabbit IgG was used as a negative control (Figs 2F and S4E). As a positive control, we analysed MLL1 occupancy at the constitutively active *GAPDH* promoter, where MLL1 levels remained unchanged across all time points (Fig S5C and D). As a negative control, we analysed MLL1 enrichment at the muscle-specific gene *MYOD1*. Because *MYOD1* is constitutively

repressed in myeloid monocytes, it exhibited only low-level background enrichment, confirming the specificity of our assay (Fig S5E and F). Collectively, these data reveal a temporal correlation between MLL1 recruitment and H3K4me3 enrichment, supporting the conclusion that H3K4me3 is deposited post-transcriptionally at the inducible gene loci. The delayed recruitment of MLL1 suggests a regulatory mechanism in which MLL1 is engaged after transcription initiation, potentially promoting H3K4me3-mediated post-transcriptional chromatin remodelling.

### Transcription is a prerequisite for H3K4me3 deposition

Although H3K4me3 is enriched at active promoters, our data suggest that its accumulation occurs post-transcriptionally at the *TNF-α* and *IL-1β* loci. We therefore hypothesized that its deposition may nonetheless depend on transcriptional activity. To evaluate this, we employed actinomycin D (ActD), a potent transcriptional inhibitor that intercalates into DNA and disrupts RNA polymerase function (Sobell, 1985). THP-1 cells were stimulated with LPS in the presence or absence of actinomycin D, and analysed at 2 and 6 h post-stimulation for *TNF-α* and *IL-1β* transcription induction and H3K4me3 enrichment at their respective promoters. LPS stimulation of naïve THP-1 cells (Naïve-LPS) induced *TNF-α* mRNA expression at 2 h, consistent with a transient inflammatory response, which diminished by 6 h (Fig 3A). In contrast, actinomycin D–treated cells stimulated with LPS (ActD-LPS) exhibited a marked reduction in *TNF-α* mRNA levels at 2 h compared with Naïve-LPS cells, indicating effective transcriptional inhibition (Fig 3A). By 6 h, *TNF-α* mRNA levels in ActD-LPS cells remained low, similar to those in Naïve-LPS cells. Consistent with the H3K4me3 dynamics observed above, H3K4me3 enrichment at the *TNF-α* promoter in Naïve-LPS cells was markedly increased at 6 h, but not at 2 h post-stimulation (Fig 3B). Interestingly, ActD-LPS cells exhibited significantly reduced H3K4me3 enrichment at 6 h compared with Naïve-LPS cells, suggesting that transcriptional inhibition impairs H3K4me3 deposition (Fig 3B). The validity of the ChIP results was monitored using an IgG control (Fig 3C). A similar effect was observed at the *IL-1β* promoter, where transcriptional blockade at 2 h impaired H3K4me3 enrichment by 6 h (Fig S6A–C). Together, these findings indicate that active transcription is necessary for the deposition of H3K4me3 at these inducible gene promoters.

In addition to the global transcription inhibitory assay, we also performed a more targeted strategy to assess the transcriptional dependency of H3K4me3 loading. For this, we used TPCA-1, a selective inhibitor of IKK2 (IκB kinase-2) that blocks NF-κB activation. As NF-κB is a central transcriptional regulator of inflammatory genes, including *TNF-α* and *IL-1β*, TPCA-1 treatment allowed us to specifically inhibit the transcription of these genes, thereby providing a gene-specific means to examine the requirement of transcription for H3K4me3 deposition (Wang et al, 2021). THP-1 cells were stimulated with LPS in the presence or absence of TPCA-1, and we examined transcriptional and dynamic H3K4me3 loading responses at 0, 2, and 6 h post-stimulation. Naïve THP-1 cells stimulated with LPS (Naïve-LPS) exhibited the expected up-regulation of *TNF-α* and *IL-1β* mRNA at 2 h, followed by a decline at 6 h (Figs 3D and S6D). In contrast, TPCA-1–treated cells (TPCA-1 LPS) displayed markedly reduced *TNF-α* and *IL-1β* mRNA levels at 2 h upon LPS stimulation, indicating effective inhibition of NF-κB–dependent

transcription (Figs 3D and S6D). By 6 h, *TNF-α* and *IL-1β* mRNA levels in TPCA-1 LPS cells remained low, similar to those observed in Naïve-LPS cells. Corroborating the transcriptional profiles, the analysis of H3K4me3 dynamics in these cells mirrored the expression patterns. In Naïve-LPS cells, H3K4me3 enrichment at the *TNF-α* promoter was minimal at 2 h but significantly elevated by 6 h (Fig 3E). However, upon TPCA-1 treatment, H3K4me3 enrichment at 6 h was strongly attenuated (Fig 3E), in line with transcriptional inhibition and as observed with ActD. IgG was used as a negative control to evaluate non-specific binding in the ChIP assay (Fig 3F). A similar H3K4me3 pattern was observed at the *IL-1β* promoter, where early NF-κB inhibition (2 h) resulted in reduced H3K4me3 levels at 6 h (Fig S6D–F). Together, these findings demonstrate that prior transcriptional activity is a prerequisite for H3K4me3 deposition.

### Transcriptional activation occurs independently of maximal H3K4me3 levels at inducible genes

Although H3K4me3 has been shown to be non-essential for transcription initiation (Wang et al, 2012; Kimball et al, 2017; Wang et al, 2024), the molecular basis for this observation has remained unclear. Here, we show that H3K4me3 is deposited after the peak of transcriptional activity, explaining its non-essential role in transcription initiation. To demonstrate this dispensability in the context of the current inducible gene activation model, we analysed the transcriptional activation of *TNF-α* under conditions of reduced H3K4me3 levels. We used shRNA to deplete MLL1, a histone methyltransferase recruited to NF-κB–bound promoters and implicated in inflammatory gene activation in monocytes and macrophages (Wang et al, 2012; Kimball et al, 2017; Wang et al, 2024). Stable THP-1 monocytic cell lines expressing shRNAs (n = 2) targeting human *MLL1* were generated using the pLKO.1 lentiviral vector system. Western blot analysis of all the H3K4 methylations confirmed efficient depletion of MLL1, accompanied by a marked global reduction in H3K4me3 levels compared with control cells transduced with the empty vector (Fig 4A and B). However, the global H3K4me1 levels did not change significantly, whereas H3K4me2 levels reduced minimally. Upon LPS stimulation, MLL1 knockdown cells exhibited *TNF-α* and *IL-1β* mRNA levels comparable to those of control cells at 2 h (Figs 4C and S7A), indicating that reduced global H3K4me3 due to MLL1 knockdown does not impair *TNF-α* and *IL-1β* induction. To confirm the status of the epigenetic landscape at these genes, H3K4me3 levels were assessed at both the *TNF-α* and *IL-1β* promoters by ChIP-qPCR. Consistent with the global reduction observed by Western blot, a significant depletion of H3K4me3 at 6 h post-transcriptional peak was detected at both promoters in the MLL1 knockdown cells compared with the control pLKO.1 cells (Figs 4D and S7B). Although a residual level of H3K4me3 remained detectable in the knockdown lines, this substantial reduction in promoter-associated density did not impair mRNA induction at the 2 h transcriptional peak. ChIP assays using non-specific IgG were conducted to evaluate the background noise (Figs 4E and S7C). Overall, these data indicate a lack of quantitative coupling between H3K4me3 density and transcriptional initiation, suggesting that high-density enrichment of this mark is not a rate-limiting prerequisite for the initiation of *TNF-α* and *IL-1β* transcription.

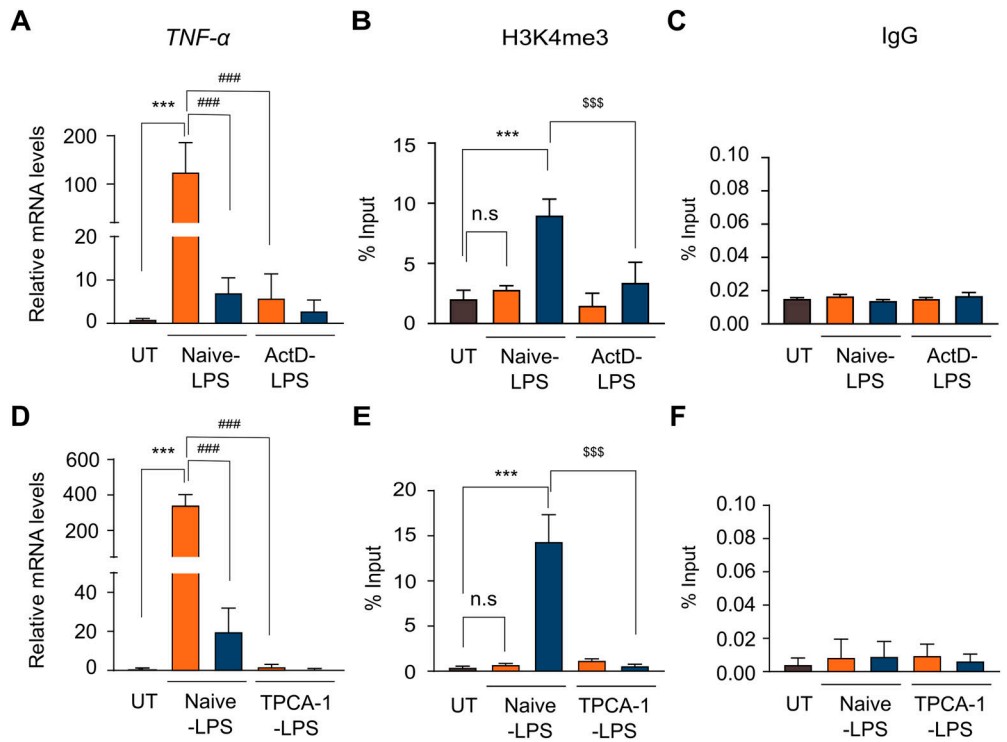

**Figure 3. Post-transcriptional H3K4me3 loading requires active transcription.**
To determine whether active transcription is necessary for the post-transcriptional deposition of H3K4me3 at the *TNF-α* promoter, THP-1 cells were pre-treated with either the transcriptional inhibitor actinomycin D (1 µM, 30 min) or the IKK2 inhibitor TPCA-1 (5 µM, 30 min) before stimulation with LPS (500 ng/ml), and harvested at 0, 2, and 6 h post-stimulation. **(A, D)** qRT-PCR analysis of *TNF-α* mRNA levels in actinomycin D–treated (A) or TPCA-1–treated (D) cells compared with naïve cells after LPS stimulation. **(B, E)** ChIP analysis using antibodies against H3K4me3 to assess enrichment at the *TNF-α* promoter in actinomycin D–treated (B) or TPCA-1–treated (E) cells. **(C, F)** ChIP with normal IgG was included as a negative control for actinomycin D (C) and TPCA-1 (F) conditions. Statistical analysis was performed using one-way ANOVA followed by Dunnett's multiple comparisons test. Data are presented as the mean ± SD from n = 3 independent biological replicates. Statistical significance is denoted as ***$P < 0.001$, **$P < 0.01$, and *$P < 0.05$ versus the unstimulated control; ###$P < 0.001$, ##$P < 0.01$, and #$P <$
0.05 compared with LPS-stimulated cells at 2 h; $$$$P < 0.001$, $$$P < 0.01$, and $$P < 0.05$ compared with LPS-stimulated cells at 6 h; and ns as non-significant compared with the unstimulated control.

## Temporal uncoupling of transcription and H3K4me3 occupancy at the *MYC* locus

Given that post-transcriptional H3K4me3 deposition was observed at inducible genes, we next investigated whether this feature is also characteristic of constitutively expressed genes. However, studying delayed transcriptional effects, such as H3K4me3 deposition, at constitutively active genes is inherently challenging because continuous transcription can mask subtle temporal dynamics. To overcome this limitation, we used transcriptional termination as a model system to investigate the dynamics of H3K4me3 deposition. We hypothesized that if H3K4me3 was deposited post-transcriptionally with a delay after transcriptional activity, it would remain at the promoter for an extended period even after transcription had terminated (Fig 5A). We selected the *MYC* gene for this analysis because it is constitutively expressed and, importantly, has a very short mRNA half-life (Herrick & Ross, 1994). The short half-life of *MYC* mRNA makes it a sensitive and reliable indicator of transcriptional status, as residual transcript levels after actinomycin D treatment closely reflect the degree of ongoing transcription. To assess transcriptional dynamics and epigenetic changes, THP-1 cells were exposed to actinomycin D for 0, 1, 2, 4, 6, and 8 h, followed by quantification of *MYC* mRNA and H3K4me3 levels at the *MYC* promoter. After actinomycin D treatment, *MYC* mRNA levels declined to ~20% by 2 h, less than 10% by 4 h, and below 5% at 6 and 8 h, compared with untreated cells at 0 h, demonstrating effective transcriptional inhibition (Fig 5B). ChIP enrichment analysis of H3K4me3 at the *MYC*

promoter revealed sustained levels up to 4 h of treatment, comparable to those observed in the untreated control (Fig 5C). Notably, its levels declined significantly at 6 and 8 h (Fig 5C). Non-specific binding was accounted by using species-matched IgG as a negative control, which consistently yielded negligible enrichment (Fig 5D). The sustained H3K4me3 levels at 2 and 4 h, which persist despite effective transcriptional shutdown, demonstrate a temporal uncoupling of transcription and H3K4me3 occupancy at the *MYC* promoter. These observations define a post-transcriptional window at the *MYC* promoter where H3K4me3 occupancy persists independently of active transcriptional activity.

## Discussion

H3K4me3 has long been presumed to facilitate transcription initiation, yet accumulating evidence indicates that it can be dispensable, complicating its mechanistic interpretation (Howe et al, 2017; Talbert & Henikoff, 2021; Yu & Lesch, 2024; Wang & Helin, 2025). This raises a fundamental question of causality: whether H3K4me3 actively promotes transcription or whether it simply marks transcriptional activity. One way to resolve this question is by dynamically analysing the timing of H3K4me3 enrichment relative to transcription. To this end, we employed two inducible human gene models (*TNF-α* and *IL-1β*) and performed a temporal dynamic analysis of transcriptional activity and histone modifications at their native gene loci. We found that H3K4me3 is not

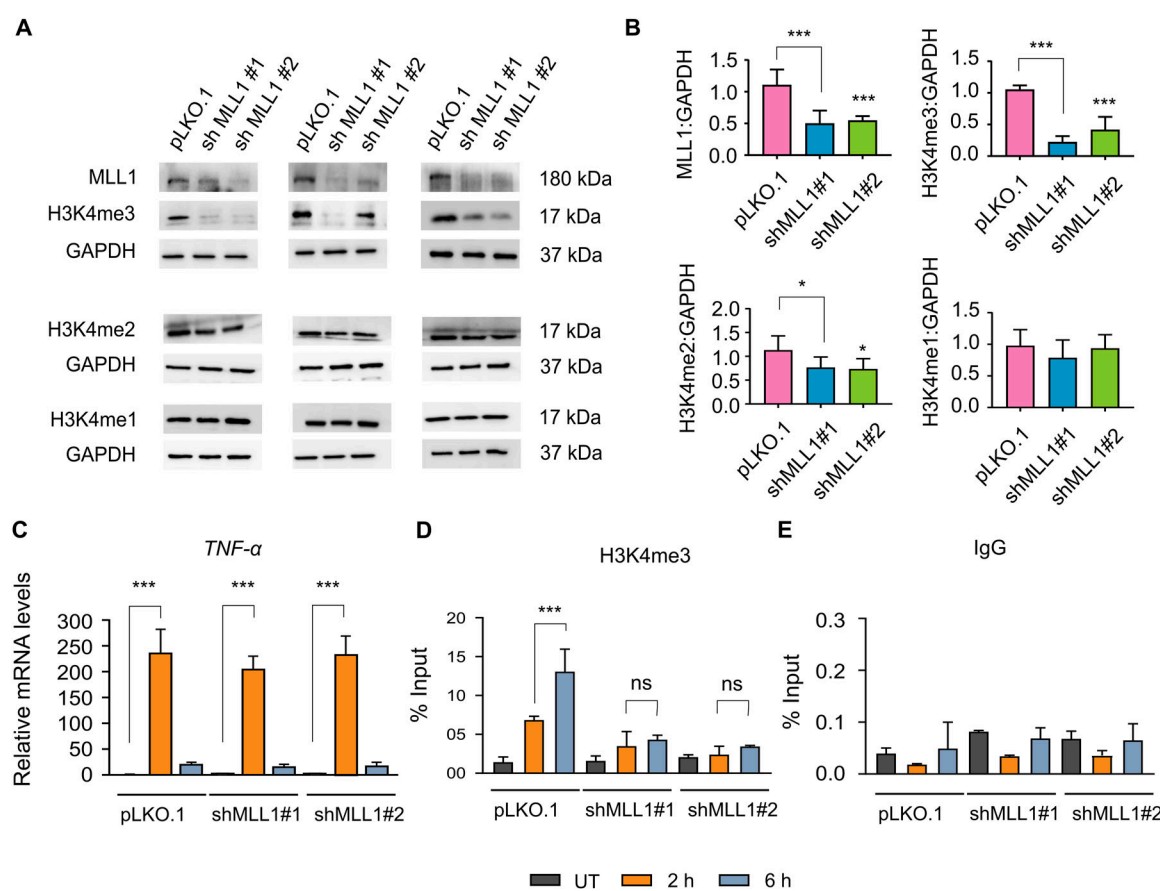

**Figure 4. H3K4me3 depletion does not impair LPS-induced *TNF-α* mRNA expression.**
THP-1 cells with MLL1 knockdown were generated by lentiviral transduction using pLKO.1 vectors expressing two independent shRNAs. Transduced cells were selected with puromycin (1 µg/ml) for 7 d to establish stable cell populations. After selection, cells were used for LPS-induced transcriptional analysis of *TNF-α*. **(A)** Representative Western blot analysis of MLL1, H3K4me1, H3K4me2, and H3K4me3 levels in control and shMLL1 cells with GAPDH as a loading control; the three sets of blots shown correspond to three independent biological replicates used for quantification. **(B)** Densitometric quantification of immunoblot signals was performed using ImageJ. **(C)** qRT–PCR analysis of *TNF-α* mRNA expression in control and shMLL1 cells after LPS stimulation (2 and 6 h). **(D, E)** ChIP-qPCR analysis showing H3K4me3 enrichment at the *TNF-α* promoter in control and shMLL1 cells after LPS stimulation, and (E) IgG was used as a negative control using the *TNF-α* promoter–specific primer. Statistical analysis was performed using one-way ANOVA with Dunnett's multiple comparisons test. Data are presented as the mean ± SD from three independent biological replicates (n = 3). Statistical significance is indicated as **$P < 0.001$, *$P < 0.01$, ###$P < 0.001$, ##$P < 0.01$, and #$P < 0.05$ compared with LPS-stimulated cells at 2 h; \$\$\$$P < 0.001$, \$\$$P < 0.01$, and \$$P < 0.05$ compared with LPS-stimulated cells at 6 h; and ns as non-significant compared with the unstimulated control. Source data are available for this figure.

enriched at promoters during peak transcription of inducible genes (*TNF-α* and *IL-1β*), indicating that this histone modification is not required for transcriptional initiation. Instead, its accumulation occurs post-transcriptionally, peaking several hours after transcription has subsided, and is dependent on prior RNA synthesis. These findings suggest that at these inducible loci, H3K4me3 acts as a consequence of transcriptional activity rather than a prerequisite for the initial induction of these genes.

In this study, we strategically employed the human inflammatory genes *TNF-α* and *IL-1β* as transcriptional models. Their well-characterized, rapid, and transient expression kinetics, coupled with short mRNA half-lives (Suzuki et al, 2000; Sharif et al, 2007; Adamik et al, 2013; Pulugulla et al, 2016), provides a high-fidelity "off–on–off" system. This temporal precision allows for a granular examination of both early initiation and delayed epigenetic effects, representing transitions that are typically obscured in constitutively expressed or slowly induced loci. By studying these genes at

their endogenous loci, we preserved native chromatin architecture and avoided the potential artefacts associated with transgene-based systems. Notably, although these genes are located on different genomic locations on different chromosomes (Chr 6 and Chr 2), they exhibited identical H3K4me3 loading kinetics. This consistency across biologically independent genomic environments suggests that the observed delay is not a site-specific outlier effect, but rather a conserved feature of rapid gene induction. Collectively, these features demonstrate that these loci are ideal models to dissect the temporal dynamics of H3K4me3, revealing it to be a consequence of transcriptional activity rather than a prerequisite for its onset.

Comparative analyses of additional histone modifications and transcriptional regulators further support the post-transcriptional deposition of H3K4me3 at these loci. Histone acetylation marks are well-established correlates of active transcription; consistent with this, enrichment profiles of

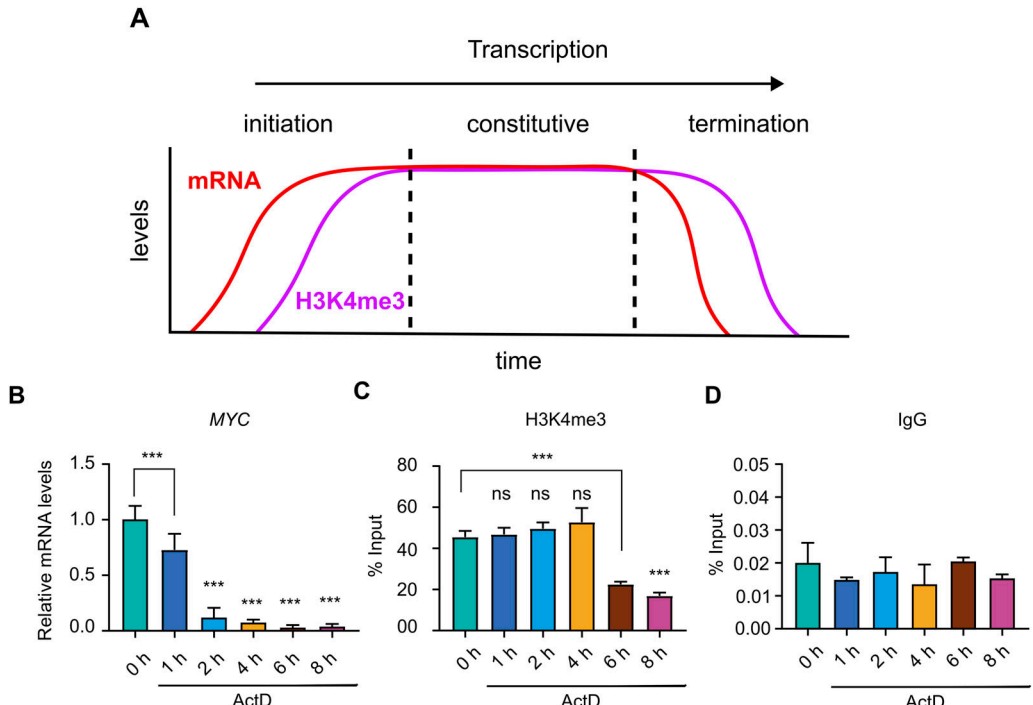

**Figure 5. Sustained H3K4me3 enrichment at the constitutively active *MYC* promoter despite transcriptional inhibition suggests delayed deposition.**
**(A)** Schematic representation illustrating the hypothesized deposition of H3K4me3 at the TSS during distinct transcriptional phases, including initiation, constitutive expression, and termination. To assess whether post-transcriptional delay in H3K4me3 deposition occurs at the constitutively active *MYC* promoter, THP-1 cells were treated with actinomycin D (1 μM) for the indicated durations (0, 1, 2, 4, 6, and 8 h), and *MYC* transcription levels and H3K4me3 enrichment at the *MYC* promoter were analysed. **(B)** qRT–PCR analysis of *MYC* mRNA levels in THP-1 cells treated with actinomycin D (1 μM) for the indicated durations (0, 1, 2, 4, 6, and 8 h), demonstrating effective transcriptional inhibition. **(C)** ChIP assays were performed using antibodies against H3K4me3 to assess its enrichment at the *MYC* promoter under transcriptionally repressed conditions. **(D)** ChIP with isotype-matched IgG was included as a negative control. Statistical analysis was performed using one-way ANOVA followed by Dunnett's multiple comparisons test. Data are presented as the mean ± SD from three independent biological replicates (n = 3). Statistical significance is denoted as \*\*\**P* < 0.001, \*\**P* < 0.01, and \**P* < 0.05 compared with the unstimulated control; and ns as non-significant compared with the unstimulated control.

H3K27ac and H4K8ac closely mirrored transcriptional activity at the *TNF-α* and *IL-1β* promoters. This observation places H3K4me3 accumulation downstream of acetylation and is consistent with previous findings, showing that targeted H3K27ac can induce H3K4me3, whereas H3K4me3 recruitment alone is insufficient to initiate transcription (Zhao et al, 2021). Although H3K4me3 enrichment was significantly lagged, the recruitment of NF-κB, p300, and phosphorylated RNA polymerase II (Ser5 and Ser2) occurred in tandem with the transcriptional peak, highlighting a distinct kinetic separation between these early activators and the subsequent loading of H3K4me3 (Ding et al, 2012). Moreover, dynamic enrichment of MLL1, the methyltransferase responsible for H3K4me2/3 (Dou et al, 2006), implicated in NF-κB–mediated transcription (Wang et al, 2012; Kimball et al, 2017; Wang et al, 2024), correlated with the delayed enrichment of H3K4me2/3 at *TNF-α* and *IL-1β* promoters. Although H3K4me3 was deposited independently of ongoing transcription, its accumulation was significantly reduced upon inhibition of *TNF-α* and *IL-1β* transcription by a specific inhibitor of NF-κB signalling (TPCA-1), as well as global transcriptional blockade using actinomycin D, indicating that prior transcriptional activity is necessary for its establishment. Evidence suggests that H3K4me3 establishment is coupled to active transcription rather than direct recruitment by NF-κB. This is supported by the observed temporal lag between rapid NF-κB binding and the

subsequent peak of H3K4me3 (Figs 1E and 2C). If NF-κB functioned as a direct physical scaffold for MLL1, these events would be expected to coincide. Furthermore, the reduction of H3K4me3 after both NF-κB inhibition and actinomycin D treatment confirms that the mark is dependent on the active transcriptional process. Collectively, these observations are consistent with a model where NF-κB initiates a transcriptional state that matures into an H3K4me3-enriched signature only after initiation, providing evidence for a post-transcriptional mechanism of epigenetic reinforcement at these inducible human genes.

Interestingly, we noted a modest increase in H3K4me1 at early time points preceding robust transcriptional activation at these gene loci (Figs 1C and S2B). This suggests an early chromatin remodelling or priming event at the promoter. We interpret this early enrichment as an intermediate chromatin configuration that precedes full gene activation. As transcription becomes established, this mark may subsequently transition-or "mature"-into H3K4me3, which is typically associated with active promoters. Although the present study does not directly examine the enzymatic conversion between these states, the observed temporal pattern is highly consistent with a stepwise model of H3K4 methylation during transcriptional activation.

The delayed deposition of H3K4me3 that we observe in our human cell models is consistent with prior findings in other systems. In budding yeast, Kuang et al (2014) reported that histone methylation marks, including H3K4me3, lag behind transcriptional activation during metabolic cycles. Similarly, in mouse liver, Le Martelot et al (2012) showed that H3K4me3 at promoters peaks after RNA polymerase II recruitment and persists even when transcription declines. These studies reinforce the broader relevance of our findings, demonstrating that the delayed H3K4me3 deposition characterized here in human inflammatory genes may represent a conserved feature across species. Beyond inducible models, we examined whether post-transcriptional H3K4me3 deposition is a general feature of constitutively active genes. However, dissecting this relationship is complicated by the continuous transcription and stable H3K4me3 levels, making it challenging to distinguish whether chromatin changes are a cause or a consequence of transcription, particularly when they occur after a temporal lag. To address this, we used a transcriptional termination model to test whether H3K4me3 is maintained at promoters after transcriptional shutdown. Using the constitutively active *MYC* gene as a model, transcriptional shutdown resulted in a rapid decline in mRNA levels, yet H3K4me3 occupancy at the promoter remained stable for at least 4 h. This observation demonstrates a clear uncoupling between active transcription and the histone methylation mark at this locus. However, we cannot rule out that this persistence may reflect either a delayed deposition or a low turnover rate of the modification; in either case, the data indicate that H3K4me3 levels do not strictly mirror transcriptional kinetics in real time. Consequently, the presence of H3K4me3 at a gene lacking active transcription may serve as an epigenetic signature of prior transcriptional activity, rather than a marker of concurrent mRNA synthesis (Guenther et al, 2007). These findings are consistent with previous reports of sustained H3K4me3 after transcriptional arrest (Ng et al, 2003), supporting a model in which this mark serves as an epigenetic record of prior transcriptional activity rather than a real-time indicator of active transcription.

Our findings add to a growing body of evidence that challenges the canonical view of H3K4me3 as a requisite mark for transcription initiation. Despite its strong enrichment at active promoters, multiple studies indicate that H3K4me3 is dispensable for the onset of transcription. In mouse embryonic stem cells, deletion of CFP1 significantly reduces H3K4me3 at CpG island-associated promoters without affecting nascent or steady-state RNA levels (Carlone et al, 2005; Clouaire et al, 2012). Similarly, *Drosophila* cells lacking H3K4me3 maintain inducible gene expression, and yeast mutants devoid of H3K4 methylation remain viable and transcriptionally competent (Hödl & Basler, 2012; Soares, 2012). Transcriptional studies using reconstituted in vitro systems demonstrated that transcription is independent of enzymes responsible for catalysing H3K4me3 (Pavri et al, 2006). Consistent with these findings, our inducible transcriptional model showed that MLL1 knockdown, which reduces H3K4me3 levels, does not impair *TNF-α* and *IL-1β* activation. Our findings in MLL1-depleted cells demonstrate a lack of quantitative coupling between H3K4me3 density and transcriptional output. Although residual levels of the mark persist in the MLL1-depleted cells, the observation that a substantial reduction in

H3K4me3 failed to impair the induction of *TNF-α* and *IL-1β* suggests the mark is not a rate-limiting prerequisite for initiation. If H3K4me3 functioned as an inductive trigger, even partial depletion should dampen the transcriptional response. Instead, achieving full mRNA induction despite significant loss of the mark supports a model where high-density H3K4me3 serves as a stable readout of sustained activity rather than a requirement for the inflammatory program to begin. Our global histone analysis reveals that although MLL1 is strictly required for the accumulation of H3K4me3, it remains dispensable for the maintenance of H3K4me1 and H3K4me2 levels. This observation aligns with established models suggesting that MLL3/4 complexes primarily manage the early deposition of "priming" marks, whereas MLL1 is specifically recruited to regulate the maturation of promoter-proximal trimethylation during active transcriptional induction (Herz et al, 2012; Calo, 2013). This study offers a plausible mechanistic explanation for the apparent disconnect between H3K4me3 enrichment at promoters and its non-essential role in transcription initiation. A recent study by Wang et al (2023) supports this perspective, showing that depletion of H3K4me3 leads to only modest global changes in gene expression and is not required for RNA polymerase II loading at promoters. However, they observed increased RNA polymerase II pausing, suggesting that H3K4me3 regulates pause release. In contrast, our data demonstrate that H3K4me3 enrichment occurs after the initial transcriptional burst, indicating that its deposition is temporally uncoupled from concurrent transcription in the inducible gene systems analysed. Together, these findings support a model where H3K4me3 functions primarily as a molecular record of transcriptional history, rather than a driver of initiation.

Although our study defines the temporal uncoupling of H3K4me3 from active transcription, the specific downstream mechanistic consequences of this post-transcriptional loading remain to be fully elucidated. Based on current literature and our observations at these specific loci, we propose that H3K4me3 may serve several post-initiation functions: (i) stabilizing promoter-proximal open chromatin to potentially act as a barrier against the encroachment of repressive histone modifications (Wysocka et al, 2006; Ma et al, 2011); (ii) facilitating transcriptional elongation or modulating RNA turnover through interactions with the transcriptional machinery (Ding et al, 2012; Hu et al, 2023; Wang et al, 2023); (iii) contributing to the maintenance of genome integrity (Gong et al, 2017; Abu-Zhayia et al, 2022); and (iv) supporting transcriptional memory (Ng et al, 2003). To fully elucidate the prevalence of these H3K4me3 dynamics and define their broader mechanistic roles, further investigation is required. Although our study was restricted to a specific subset of gene models, this detailed kinetic analysis provides evidence for a post-transcriptional role that remains largely uncharacterized in steady-state studies. Moving forward, expanding these investigations to a genome-wide scale will be essential to determine whether the temporal uncoupling observed here is a feature of inducible gene models or a broader property of the epigenome. However, investigating these dynamics at constitutively active genes-which comprise most of the genome-is challenging, as ongoing transcription masks the underlying kinetics of probable delays in H3K4me3 deposition. This necessitates specialized approaches, such as the transcriptional inhibition-recovery assays employed here or metabolic histone labelling. Ultimately, these analyses will provide a more comprehensive

understanding of the functional significance and regulatory contribution of H3K4me3 throughout the transcriptional cycle.

In conclusion, our findings characterize H3K4me3 as a transcription-dependent, post-transcriptional epigenetic mark at the human inflammatory genes studied. Rather than serving as a prerequisite for initiation, H3K4me3 enrichment reflects prior transcriptional activity. By defining the temporal uncoupling between transcription and H3K4me3 at inducible and constitutive loci, our study contributes to the understanding of this modification as a record of transcriptional history. These observations provide a dynamic framework to evaluate the potential post-transcriptional role of H3K4me3 at other genomic sites. Ultimately, this work reinforces a model of H3K4me3 function where the mark acts as a key player in the epigenetic processing that follows transcription, rather than serving as a driver of initiation.

# Materials and Methods

## Cell culture

THP-1 monocytic cells (ATCC) were cultured in suspension in RPMI-1640 medium (Cat# 23-400-021; Gibco) supplemented with 10% heat-inactivated FBS (Cat# 10270-106; Gibco) and 1% of penicillin–streptomycin (Cat# 15140-122; Gibco). Cells were maintained in a humidified incubator at 37°C with 5% $CO_2$ and subcultured every 2–3 d. For transcriptional activation studies, THP-1 monocytes ($1 \times 10^7$ cells) were cultured in 10 ml of complete RPMI medium and induced with *E. coli* LPS (500 ng/ml) (Cat# L4005; Sigma-Aldrich) for various durations (0, 0.5, 2, 6, 12, and 24 h). At each time point, 1 ml of the culture was harvested for RNA isolation, whereas the remaining 9 ml was processed for chromatin immunoprecipitation (ChIP) assays. For lentiviral production, HEK293T cells were cultured in DMEM (Cat# 12100-061; Gibco) supplemented with 10% FBS and 1% penicillin–streptomycin solution. Cells were cultured under the same conditions as THP-1 cells.

## Lentivirus plasmid construction and transfection

shRNA sequences targeting human *MLL1* were designed based on the Broad Institute's TRC (Transgenic RNAi Consortium) shRNA design process (Moffat et al, 2006), and oligonucleotides were synthesized and annealed (listed in Table S1). The resulting duplexes were ligated into the pLKO.1- vector (Addgene# 8453), linearized with *Age* I and *EcoR* I, using the Quick T4 DNA Ligase (Cat# M2200L; NEB) kit. HEK293T cells were transfected with either sh*MLL1* or an empty vector (control) for lentiviral production along with third-generation lenti-packaging plasmids pREV, pRRE, and pMD2.G using Xfect Transfection Reagent (Cat# 631318; Takara) based on the manufacturer's protocol. After 48 and 72 h of post-transfection, viral particles were collected. Lentiviral transduction was performed on THP-1 cells ($5 \times 10^5$ cells/ml) in the presence of 8 $\mu$g/ml polybrene (Cat# 107689; Sigma-Aldrich). After 24 h, cells were added with fresh RPMI-1640 medium, and puromycin (Cat# P4512; Sigma-Aldrich) selection was carried out with 1 $\mu$g/ml

concentration for 7 d. Later, MLL1 knockdown efficiencies were evaluated by Western blot analysis, and subsequently, these cells were used to analyse LPS-induced activation of *TNF-α* transcription.

## RNA extraction, cDNA synthesis, and qRT–PCR

Total RNA was isolated using RNAiso Plus (Cat# 9109; Takara) according to the manufacturer's instructions. PrimeScript RT Reagent Kit (Cat# 6110A; Takara) was used to prepare cDNA from 1 $\mu$g of total RNA. Quantitative reverse transcription PCR (qRT-PCR) was performed with Takara SYBR Green Master Mix (Cat# RR820) using gene-specific primers (Table S2) on Applied Biosystems 7900HT Real-Time PCR System. mRNA expression levels were normalized to *β2-microglobulin* (*β2M*) as a housekeeping control, and relative quantification was calculated using the $\Delta\Delta C_t$ method. Statistical analysis was performed by GraphPad Prism v.9 software (GraphPad).

## Chromatin immunoprecipitation (ChIP)

ChIP assays were performed as previously described (Ansalone et al, 2021). Briefly, THP-1 cells were fixed with 1% formaldehyde for 10 min at room temperature (RT), and crosslinking was quenched with 0.125 M glycine. Chromatin was prepared by lysing the cells in ChIP lysis buffer (20 mM Hepes, pH 7.6, 1% SDS, 1 mM sodium butyrate, protease inhibitors) on ice. Chromatin was sonicated using a bath sonicator (SONOREX) to yield DNA fragments of ~200–600 bp. The lysate was pre-cleared by centrifugation, and 10% of the input was reserved for normalization. For each ChIP reaction, 20 $\mu$l of Protein A Dynabeads (Cat# 10002D; Invitrogen) was coated with 1.5 $\mu$g of the respective antibody (listed in Table S3), diluted in 0.1% BSA-containing ChIP dilution buffer (16.7 mM Tris–HCl [pH 8.0], 1.2 mM EDTA, 167 mM NaCl, 1% Triton X-100, 0.01% SDS), and incubated for 2 h at RT with gentle rotation. The antibody-coated beads were then added to the pre-cleared chromatin (equivalent to $1 \times 10^6$ cells per sample) in ChIP dilution buffer and incubated at 4°C overnight with rotation. The next day, magnetic beads containing the immunoprecipitated chromatin complexes were separated using a magnetic stand. Beads were then sequentially washed with ice-cold buffers: once with low-salt wash buffer (20 mM Tris–HCl, pH 8.0, 2 mM EDTA, 150 mM NaCl, 0.1% SDS, 1% Triton X-100), twice with high-salt wash buffer (20 mM Tris–HCl, pH 8.0, 2 mM EDTA, 500 mM NaCl, 0.1% SDS, 1% Triton X-100), and twice with TE buffer (10 mM Tris–HCl, pH 8.0, 1 mM EDTA). Chromatin was eluted in 100 $\mu$l elution buffer (10 mM Tris–HCl, pH 8.0, 5 mM EDTA, 300 mM NaCl, 0.5% SDS) supplemented with 2 $\mu$l proteinase K (20 mg/ml) and incubated overnight at 65°C for de-crosslinking the chromatin. ChIP DNA was purified using PCR Clean-up/Gel Extraction Kit (Cat# NP-36107; Genetix). Enrichment of target genomic regions was analysed by quantitative real-time PCR (qPCR) using promoter-specific primers listed in Table S2. ChIP enrichment was calculated using the $\Delta Ct$ method and expressed as a percentage of the input DNA (% input).

### Transcription inhibition analysis using actinomycin D and TPCA-1

To investigate the transcription dependence of chromatin modifications, THP-1 cells (1 × 10^6 cells per time point) were pre-treated with 1 $\mu$M actinomycin D (Cat# A4262; Sigma-Aldrich) or 5 $\mu$M of TPCA-1 (Cat# T1452; Sigma-Aldrich) for 30 min to inhibit transcription. After pre-treatment, cells were induced with LPS (500 ng/ml) for 0, 2, and 6 h and harvested for downstream analysis of *TNF-α* and *IL-1β* mRNA expression by qRT–PCR and for H3K4me3 ChIP enrichment at the *TNF-α* and *IL-1β* promoters, as described. Untreated naïve cells subjected to identical LPS stimulation served as experimental controls.

To evaluate whether the post-transcriptional accumulation of H3K4me3 represents a general chromatin regulatory mechanism (Fig 5), THP-1 cells (1 × 10^6 per ml) were treated with 1 $\mu$M actinomycin D (Cat# A4262; Sigma-Aldrich) for varying durations (0, 1, 2, 4, 6, and 8 h). Cells were harvested at each time point for downstream analysis of *MYC* mRNA expression by qRT–PCR and for H3K4me3 chromatin immunoprecipitation (ChIP) enrichment at the *MYC* promoter by ChIP-qPCR. Untreated cells (0 h) served as the control condition.

### Western blot

Total proteins were isolated by lysing cells in cell lysis buffer (Cat# ST0349; Takara) supplemented with Halt Protease Inhibitor Cocktail (Cat# 635673; Takara) and Phosphatase Inhibitor Cocktail (Cat# P0044; Sigma-Aldrich). Protein concentration was estimated using BCA Protein Assay Kit (Cat# 71285-3; Thermo Fisher Scientific). Equal amounts of protein (30 $\mu$g per sample) were resolved by SDS–PAGE on 15% and 8% acrylamide gels (Bio-Rad) and transferred onto PVDF membranes (Cat# 10600023; Amersham Hybond). Membranes after protein transfer were blocked with 5% BSA with 1× TBST (137 mM NaCl, 2.7 mM KCl, 20 mM Tris–HCl [pH 7.4], 0.1% Tween-20), for 1 h at RT, and later incubated at 4°C overnight with the respective primary antibodies against GAPDH, MLL1, H3K4me1, H3K4me2, and H3K4me3 (Cat. No. listed in Table S3). After washing with 1× TBST, membranes were incubated with HRP-conjugated goat anti-rabbit IgG (Cat# AB_2313567; Jackson ImmunoResearch Laboratories) secondary antibodies. Signal was developed using the ECL Select Western Blotting Detection Kit (Cat# RPN2235; Amersham) and visualized with ChemiDoc Imaging System (Bio-Rad). Densitometric analysis of Western blot bands was performed using ImageJ software. Band intensities were normalized to GAPDH to account for variations in protein loading. Original blots are provided in source data file.

### Statistical significance

Statistical analysis for qRT–PCR and ChIP-qPCR enrichment assays was performed using one-way ANOVA with Dunnett's multiple comparisons test using GraphPad Prism v9.0. Data are presented as the mean ± SD from three independent biological replicates. Statistical significance is denoted as *$P < 0.05$, **$P < 0.001$, and ***$P < 0.0001$ compared with control cells; ###$P < 0.001$, ##$P < 0.01$, and #$P < 0.05$ compared with control LPS-stimulated cells at 2 h; $$$$$P < 0.001$, $$$P < 0.01$, and $$P < 0.05$ compared with LPS-stimulated

cells at 6 h; and ns as non-significant compared with unstimulated (control) cells. Statistical methods and data analysis are detailed in the figure legends. Analyses of Western blot in Fig 4A were independently performed three times with similar findings.

## Data Availability

All data supporting the findings of this study are present within the article and in the supplementary materials.

## Supplementary Information

## Acknowledgements

The authors thank the Director, CSIR-IICT, and Dr. Shasi Vardhan Kalivendi, HOD, Applied Biology, CSIR-IICT, Hyderabad, India, for providing the facilities necessary for conducting this research work (CSIR-IICT manuscript communication number: IICT/Pubs./2025/207). KP Walvekar acknowledges DST Inspire, Govt. of India, for providing SRF. This work was supported by the DBT-Ramalingaswami Re-entry Fellowship (fellowship no. BT/RLF/Re-entry/2012019) awarded to S Chilaka. S Chilaka acknowledges the DBT-Ramalingaswami Re-entry Fellowship, Govt. of India, New Delhi. We thank Dr. Sistla Ramakrishna and Dr. Sai Balaji Andugulapati, IICT, for support. The authors declare no competing financial interests.

### Author Contributions

KP Walvekar: data curation, formal analysis, validation, investigation, methodology, and writing—original draft.
S Chilaka: conceptualization, data curation, formal analysis, supervision, funding acquisition, investigation, methodology, project administration, and writing—original draft, review, and editing.

### Conflict of Interest Statement

The authors declare that they have no conflict of interest.

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
