## [Reviewer comments · Life Science Alliance]

Temporal analysis of two inducible human genes reveals post-transcriptional H3K4me3 deposition

Komal Walvekar and Sabarinadh Chilaka
DOI: <https://doi.org/10.26508/lsa.202503511>

Corresponding author(s): Sabarinadh Chilaka, Indian Institute of Chemical Technology

Review Timeline:

Submission Date:	2025-09-15
Editorial Decision:	2025-11-03
Revision Received:	2026-03-13
Editorial Decision:	2026-04-07
Revision Received:	2026-04-13
Accepted:	2026-04-17

Scientific Editor: Sarita Hebbar

Transaction Report:

October 29, 2025

Re: Life Science Alliance manuscript #LSA-2025-03511-T

Dr. Sabarinadh Chilaka
Indian Institute of Chemical Technology
Applied Biology
Tarnaka
Hyderabad, Telangana 500007
India

Dear Dr. Chilaka,

Thank you for submitting your manuscript entitled "Human Inducible Gene Model Identifies H3K4me3 as a Post-Transcriptional Histone Mark" to Life Science Alliance. The manuscript was assessed by three expert reviewers, whose comments are appended to this letter. The reviewers agree that this work adds new insight into the temporality for histone H3K4me3 deposition in the context of transcription. That said all the reviewers had several concerns that preclude publication at this stage.

We agree with the reviewers that the following concerns must be addressed before publication:

1. You must tone down the conclusion that this work provides a new finding that H3K4me3 is dispensable for transcriptional initiation. Further, we encourage you to follow the suggestions of the reviewers in modifying the title of the study, the description, and discussion of pertinent findings.
2. You must address all the reviewer concerns with new data. Wherever the addition of requested data is not possible, a scientific rationale and/or a caveat on interpretations must be clearly discussed in the manuscript text.
3. You must resolve the points on the Westerns/Immunoblots (Reviewer 1 and Reviewer 3, point 4).

In line with their overall assessment, we invite you to submit a revised manuscript addressing the reviewers' comments. When submitting the revision, please include a letter addressing the reviewers' comments point by point.

Thank you for this interesting contribution to Life Science Alliance. We are looking forward to receiving your revised manuscript.

Sincerely,

Sarita Hebbar, PhD
Scientific Editor
Life Science Alliance
<http://www.lsajournal.org>

B. MANUSCRIPT ORGANIZATION AND FORMATTING:

Reviewer #1 (Comments to the Authors (Required)):

In this manuscript, the authors address the following paradox: H3K4me3 is associated with active promoters but seems to be dispensable for transcription initiation. They investigate whether H3K4me3 drives transcriptional activity or whether its deposition is a consequence of transcription. To answer this question, they used inducible gene models (TNF- α and IL-1 β) in human monocytic THP-1 cells and performed time-resolved analyses of transcriptional activity and histone modifications upon inductive conditions. They showed that both H3K4me3 deposition and the enrichment of the methyltransferase MLL1 at the TNF- α and IL-1 β promoters are delayed and occur once the transcriptional activity has already decreased. Furthermore, they found that H3K4me3 is not deposited at promoters upon transcription inhibition, and that transcription is not impaired following depletion of MLL1. The authors extended their analysis to the constitutively expressed gene MYC and showed that H3K4me3 is retained at its promoter few hours after transcription shutdown. Overall, the manuscript is clearly written and easy to follow. Although previous works have shown H3K4me3 to be dispensable for transcriptional initiation, this work provides new insights on the temporality of H3K4me3 deposition. However, some points require clarification to support the authors' conclusions.

Major Comments:

- The manuscript lacks a schematic figure in the supplemental data depicting the genes studied in the paper and the targeted regions amplified for the CHIP-qPCRs and qRT-PCRs.
- Figure 1a, S1a/Lines 124-125: The authors claim that transcription levels continue to decrease at 12h and 24h but the figure doesn't include statistics supporting this decrease.
- The authors should clarify in the legend of the figure 4a what the three different blots correspond to.
- Figure 4: The immunoblot quantification only indicates a global reduction of H3K4me3 but doesn't show a loss of H3K4me3 at the TNF- α promoter specifically. To support the results shown in this figure, a H3K4me3 ChIP experiment showing the actual loss of H3K4me3 at TNF- α promoter upon MLL1 depletion should be added.
- Figure 4c: The authors should show the relative mRNA levels of IL-1 β upon depletion of MLL1 and a CHIP-qPCR showing the loss of H3K4me3 at IL-1 β promoter.
- Figure 5/line 304-305: In the result section, the authors claimed that the results showed in figure 5 demonstrates that H3K4me3 deposition is delayed compared to transcription initiation, as it was shown for inducible genes. However, the experiment only shows that H3K4me3 is retained at MYC promoter after transcriptional shutdown.
- The authors should refrain from using SEM in their figures, and show only SD instead. This allows a better estimation of the variance.

Minor comments:

- In figure 2 and S2, it would be clearer for the readers if the name of the genes corresponding to the data were added in the title of the figure or in the figure directly.
- In figure 3, the y-axis scales differ between the Fig3c and 3e (ActD and TPCA-1 input panels).
- In the legend of figure S5., it is said that the blots correspond to data shown in figure 5a, it actually corresponds to figure 4a.

Reviewer #2 (Comments to the Authors (Required)):

In this paper, the authors re-visit the role of histone H3K4me3 in transcriptional regulation. This is known to be deposited at active gene promoters and hence often implicated in the transcriptional activation process. However, there is a growing body of evidence that this mark may have a post-transcriptional role instead. Here, further evidence is gathered to support the latter hypothesis, by demonstrating that K3K4me3 deposition occurs later than transcriptional activation. This correlative observation is interesting, but the authors do not provide any data about what the role of this delayed deposition is. Overall, the study therefore adds further data to support a potential post-transcriptional role in gene regulation (or potentially no role and just a marker of previous transcription). It does not however provide a new paradigm as this observation has been made in other systems (as acknowledged in the discussion), and the authors should make this clearer in the abstract where they use the term "redefine" which would be better replaced with something like "confirm".

There are a number of specific issues that should be addressed:

- (1) In Fig.1C (and S1C), there is a slight rise in H3K4me1 that precedes transcriptional activation. This should be commented on more as might be important. It might be that this mark matures later into H3K4me3? The latter possibility could also be added to the discussion.
- (2) Fig.3d does not address transcriptional dependency as stated in the text. It assesses the role of NFkB. In that context, the role of NFkB might be to recruit MLL and this effect might itself be dependent or independent of transcriptional activity. The rationale therefore needs altering.
- (3) In Fig.4 what happens to H3K4me1/2 levels? Also, what about the levels of all H3K4me marks at enhancers surrounding these genes? What are their temporal dynamics?
- (4) I do not agree with the interpretation of Fig.5. The authors provide one explanation. However, equally plausible is that H3K4me3 turnover is slow and hence why it remains longer than transcriptional activity. Is this not direct evidence of delayed deposition. I do not think this figure adds much to the paper.
- (5) The authors state that "binding" of MLL1 does not change in Supp Fig.S3c. However, the % input is very low, so this might represent no binding at all. This needs a negative control genomic region added, to establish the basal level of antibody reactivity.
- (6) Overall, again acknowledged by the authors, this study is limited to a couple of examples which may be atypical. It would have been better to perform RNA-seq and ChIP-seq for H3K4me3 across an activation timecourse to provide more definitive results.

Reviewer #3 (Comments to the Authors (Required)):

Summary

Walvekar et al. used two inducible human genes (TNF- α and IL-1 β) to examine the timing of H3K4me3 enrichment relative to transcription. They conducted time-course profiling of transcriptional activity and histone modifications at one location downstream of the transcription start sites of TNF- α and IL-1 β . This setup shows that H3K4me2 and H3K4me3 deposition and MLL1 occupancy do not align with the activation of these genes. Instead, MLL1 occupancy and H3K4me2/3 enrichment are delayed, peaking about 6 hours after gene induction. In contrast, p300 and histone acetylation (H3K27ac and H4K8ac) follow the same pattern as mRNA transcription, peaking at 2 hours after LPS stimulation. Finally, inhibiting RNA synthesis with Actinomycin D, along with time-course profiling of transcription activity and histone modifications downstream of MYC's transcription start site, shows that H3K4me3 remains present roughly 2 hours after transcriptional inhibition. This temporal separation between transcription and deposition of H3K4me3 supports the conclusion that H3K4me3 functions as a downstream marker of active transcription rather than a prerequisite for its initiation and therefore represents a "post-transcriptional histone mark".

In summary, the authors provide another example of a conclusion that the field has already reached (1,2). In fact, the authors discuss relevant literature in their discussion. Therefore, the central claim that H3K4me3 follows transcription and depends on prior RNA synthesis, serving as a post-transcriptional histone mark rather than a driver of expression, is not new. However, there is some value in demonstrating this in a temporal manner at inducible human genes.

Comments

1. The main claim of this manuscript is that H3K4me3 follows transcription and depends on prior RNA synthesis, serving as a post-transcriptional histone mark rather than a driver of expression. In my opinion, the field has already concluded this from

studies over the last two decades. H3K4 methylation has been demonstrated to be dispensable for transcriptional activation from yeast to mouse (3-9). In the context of MLL2, H3K4me3 does not activate gene expression but instead prevents a default repressed state induced by de novo DNA methyltransferases and PRC2 activity (10). Thus, H3K4 methylation is not required for transcriptional activation and instead regulates promoter-proximal pausing through a mechanism involving the recruitment of the Integrator complex, which is essential for the eviction of paused RNAPII and transcriptional elongation (11).

Previous evidence for the deposition of H3K4me3 as a result of transcription comes from time-resolved experiments in yeast, *Drosophila*, and mouse. Research on the oscillating expression of over half of budding yeast genes during the metabolic cycle shows that, for most of these genes, H3K4me3 peaks after the maximum of their RNA transcripts (12). During meiosis in yeast, a delay in H3K4me3 compared to induced RNA is also observed (13). In zebrafish, bulk H3K4me3 is absent before the initiation of transcription during the maternal-zygotic transition (14). Finally, for genes with expression changes during diurnal cycles in the mouse liver, H3K4me3 generally peaked 1.3 hours after RNA polymerase II occupancy (15).

Mellor et al. have discussed the non-instructive role for H3K4me3 in transcription (1). I directly quote: "Another possibility is that, although H3K4me3 and transcription correlate strongly, these are actually independent events. A simple explanation is that an NDR over DNA with a suitable base composition, such as a CpG island in mammals, recruits SET1A/B leading to H3K4me3. Independently, an NDR might also be sufficient to recruit RNA polymerase II, either supported by DNA-bound transcription factors or not, leading to transcription, hence the apparent link between H3K4me3 and transcription. However, as the non-perfect correlation between H3K4me3 and transcription suggests, it is possible to get H3K4me3 without transcription and vice versa. Evidence for this comes from mESCs: at engineered CpG islands with no nearby promoters, CFP1-dependent H3K4me3 can be deposited when surrounding transcription levels, assessed using RNA polymerase II CHIP, are very low or absent and this H3K4me3 does not, in turn, lead to transcription" (16).

Lastly, the authors also acknowledge that the dynamics and regulatory roles of H3K4me3 need to be studied across a broader range of active promoters, ideally at genome-wide resolution. Based on the above comments, the central claim of the manuscript is not new, but there is some value in demonstrating it again, in a temporal manner, using two inducible human genes.

2. I recommend revising the title, "Human Inducible Gene Model Identifies H3K4me3 as a Post-Transcriptional Histone Mark," to a more suitable alternative. A new title that better reflects the scope and findings could do justice to the study without overselling the claims.

3. The authors claim to have performed "high-resolution" analyses of transcriptional activity and histone modifications in human cells. However, their analyses focus on two loci approximately 500 bp upstream of the corresponding transcription start sites (TSS). What is the reason for choosing these promoter locations? H3K4me3 shows greater enrichment at the +1 nucleosome just downstream of the nucleosome-depleted region (NDR). The authors should include at least a second region downstream of the TSS in their histone modification profiling studies.

4. In Figure 4, the authors generated stable THP-1 monocytic cell lines expressing two shRNAs targeting human MLL1 using the pLKO.1 lentiviral vector system. Western blot analysis confirmed the reduction of MLL1 by ~50%, which is partially accompanied by a global decrease in H3K4me3 levels compared to control cells transduced with the empty vector (Fig. 4a-b). The three Western blots clearly show differences in the knockdown efficiency of MLL1 (~50%-75%) and, consequently, global H3K4me3 levels (~30%-75% reduction). It is unclear to the reader whether the three western blots correspond to six independent stable THP-1 lines and which cell lines were used for densitometric quantification in Figure 4b. Is the data displayed as the mean from three independent stable THP-1 lines?

In Figure 4c, a quantitative RT-PCR analysis of TNF- α mRNA expression in control and shMLL1 cells following LPS stimulation (2 h and 6 h) is shown. Have the authors measured the effect of MLL1 knockdown on the H3K4me3 levels at the TNF- α promoter following LPS stimulation? I am missing this important experiment here to conclude that H3K4me3 depletion does not impair LPS-induced TNF- α mRNA expression.

5. The antibodies used in this study appear to have lower specificity compared to other commercial antibodies (<http://www.histoneantibodies.com/FinalArrayData/H3K4me3/>). The authors could use another H3K4me3 with higher specificity to confirm their results on delayed H3K4me3 deposition after LPS stimulation, ideally at two promoter locations (\pm 500 bp upstream and downstream of the TSS) of the TNF- α and IL-1 β genes.

6. Hypermethylated H3K4 within the mRNA coding region of yeast genes persists for a considerable time after transcription is shut down and Set1 dissociates from chromatin (17), indicating that H3K4 methylation acts as a molecular memory of recent transcriptional activity, as the authors suggest. The authors could have further investigated the potential role of H3K4 methylation as a molecular memory, but they do not. Can the authors provide additional experimental evidence that H3K4 methylation functions as a molecular memory in inducible human genes?

1. Howe FS, Fischl H, Murray SC, Mellor J. Is H3K4me3 instructive for transcription activation? *BioEssays*. 2017 Jan;39(1):1-12.
2. Morgan MAJ, Shilatifard A. Reevaluating the roles of histone-modifying enzymes and their associated chromatin modifications in transcriptional regulation. *Nat Genet*. 2020 Dec;52(12):1271-81.

3. Miller T, Krogan NJ, Dover J, Erdjument-Bromage H, Tempst P, Johnston M, et al. COMPASS: A complex of proteins associated with a trithorax-related SET domain protein. *Proc Natl Acad Sci*. 2001 Nov 6;98(23):12902-7.
4. Nislow C, Ray E, Pillus L. SET1 , A Yeast Member of the Trithorax Family, Functions in Transcriptional Silencing and Diverse Cellular Processes. *Mol Biol Cell*. 1997 Dec;8(12):2421-36.
5. Roguev A. The *Saccharomyces cerevisiae* Set1 complex includes an Ash2 homologue and methylates histone 3 lysine 4. *EMBO J*. 2001 Dec 17;20(24):7137-48.
6. Briggs SD, Bryk M, Strahl BD, Cheung WL, Davie JK, Dent SYR, et al. Histone H3 lysine 4 methylation is mediated by Set1 and required for cell growth and rDNA silencing in *Saccharomyces cerevisiae*. *Genes Dev*. 2001 Dec 15;15(24):3286-95.
7. Hödl M, Basler K. Transcription in the Absence of Histone H3.2 and H3K4 Methylation. *Curr Biol*. 2012 Dec;22(23):2253-7.
8. Denissov S, Hofemeister H, Marks H, Kranz A, Ciotta G, Singh S, et al. Mll2 is required for H3K4 trimethylation on bivalent promoters in embryonic stem cells, whereas Mll1 is redundant. *Development*. 2014 Feb 1;141(3):526-37.
9. Hu D, Garruss AS, Gao X, Morgan MA, Cook M, Smith ER, et al. The Mll2 branch of the COMPASS family regulates bivalent promoters in mouse embryonic stem cells. *Nat Struct Mol Biol*. 2013 Sept;20(9):1093-7.
10. Douillet D, Sze CC, Ryan C, Piunti A, Shah AP, Ugarenko M, et al. Uncoupling histone H3K4 trimethylation from developmental gene expression via an equilibrium of COMPASS, Polycomb and DNA methylation. *Nat Genet*. 2020 June;52(6):615-25.
11. Wang H, Fan Z, Shliaha PV, Miele M, Hendrickson RC, Jiang X, et al. H3K4me3 regulates RNA polymerase II promoter-proximal pause-release. *Nature*. 2023 Mar 9;615(7951):339-48.
12. Kuang Z, Cai L, Zhang X, Ji H, Tu BP, Boeke JD. High-temporal-resolution view of transcription and chromatin states across distinct metabolic states in budding yeast. *Nat Struct Mol Biol*. 2014 Oct;21(10):854-63.
13. Borde V, Robine N, Lin W, Bonfils S, Géli V, Nicolas A. Histone H3 lysine 4 trimethylation marks meiotic recombination initiation sites. *EMBO J*. 2009 Jan 21;28(2):99-111.
14. Vastenhouw NL, Zhang Y, Woods IG, Imam F, Regev A, Liu XS, et al. Chromatin signature of embryonic pluripotency is established during genome activation. *Nature*. 2010 Apr 8;464(7290):922-6.
15. Le Martelot G, Canella D, Symul L, Migliavacca E, Gilardi F, Liechti R, et al. Genome-Wide RNA Polymerase II Profiles and RNA Accumulation Reveal Kinetics of Transcription and Associated Epigenetic Changes During Diurnal Cycles. Kramer A, editor. *PLoS Biol*. 2012 Nov 27;10(11):e1001442.
16. Thomson JP, Skene PJ, Selfridge J, Clouaire T, Guy J, Webb S, et al. CpG islands influence chromatin structure via the CpG-binding protein Cfp1. *Nature*. 2010 Apr;464(7291):1082-6.
17. Ng HH, Robert F, Young RA, Struhl K. Targeted Recruitment of Set1 Histone Methylase by Elongating Pol II Provides a Localized Mark and Memory of Recent Transcriptional Activity. *Mol Cell*. 2003 Mar;11(3):709-19.

Authors sincerely thank the Editor-in-Chief, handling editor, and reviewers for their constructive comments and valuable suggestions to enhance our manuscript. We are also grateful for the encouraging feedback we received.

As per the LSA journal norms, we have prepared a point-by-point response to each of the reviewer's comments.

- Reviewer comments are in '*blue and italics*' and author responses are in black and plain text
- **Yellow highlights** in the manuscript represent modified text or added text as per the review comments in the revised manuscript

Current title of the manuscript: **Temporal analysis of inducible human genes reveals post-transcriptional H3K4me3 deposition**

Submission ID: **LSA-2025-03511-T**

Reviewer #1 (Comments to the Authors (Required)):
In this manuscript, the authors address the following paradox: H3K4me3 is associated with active promoters but seems to be dispensable for transcription initiation. They investigate whether H3K4me3 drives transcriptional activity or whether its deposition is a consequence of transcription. To answer this question, they used inducible gene models (TNF- α and IL-1 β) in human monocytic THP-1 cells and performed time-resolved analyses of transcriptional activity and histone modifications upon inductive conditions. They showed that both H3K4me3 deposition and the enrichment of the methyltransferase MLL1 at the TNF- α and IL-1 β promoters are delayed and occur once the transcriptional activity has already decreased. Furthermore, they found that H3K4me3 is not deposited at promoters upon transcription inhibition, and that transcription is not impaired following depletion of MLL1. The authors extended their analysis to the constitutively expressed gene MYC and showed that H3K4me3 is retained at its promoter few hours after transcription shutdown. Overall, the manuscript is clearly written and easy to follow. Although previous works have shown H3K4me3 to be dispensable for transcriptional initiation, this work provides new insights on the temporality

of H3K4me3 deposition. However, some points require clarification to support the authors' conclusions.

We thank the reviewer for their positive and constructive assessment of our manuscript. We appreciate their recognition of the novelty of our time-resolved analysis, which provides new insight into the temporal relationship between transcriptional activity and H3K4me3 deposition. We have clarified the relevant points in the revised manuscript to better support our conclusion that H3K4me3 deposition follows transcriptional activity rather than driving transcription initiation.

Major Comments:

1. The manuscript lacks a schematic figure in the supplemental data depicting the genes studied in the paper and the targeted regions amplified for the ChIP-qPCRs and qRT-PCRs.

Response:

We thank the reviewer for this helpful suggestion. A schematic figure has now been added to the Supplemental Data (Figure S1) illustrating the genes (*TNF- α* and *IL-1 β* and *MYC*) analysed in this study, along with the specific regions targeted for ChIP-qPCR and qRT-PCR amplification. This addition improves clarity regarding the experimental design and facilitates a clearer understanding of primer locations and the genomic context of the analysed regions.

2. Figure 1a, S1a/Lines 124-125: The authors claim that transcription levels continue to decrease at 12h and 24h but the figure doesn't include statistics supporting this decrease.

Response:

We thank the reviewer for pointing this out. Statistical analyses comparing transcription levels at 12 h and 24 h have now been added to Figure 1a and Figure S1a. These analyses confirm that transcription levels continue to decrease at these time points. The corresponding statistical significance and details have been included in the figure legends.

3. The authors should clarify in the legend of the figure 4a what the three different blots correspond to.

Response:

We thank the reviewer for this comment. The legend of Figure 4a has been revised to clearly

indicate that the three panels represent Western blot images from three independent biological replicates (n = 3). The legend now explicitly described this, thereby improving clarity and interpretation of the figure.

4. Figure 4: The immunoblot quantification only indicates a global reduction of H3K4me3 but doesn't show a loss of H3K4me3 at the TNF- α promoter specifically. To support the results shown in this figure, a H3K4me3 ChIP experiment showing the actual loss of H3K4me3 at TNF- α promoter upon MLL1 depletion should be added.

Response:

To address the reviewer's comment, we have expanded Figure 4 to include ChIP analysis to quantify H3K4me3 enrichment specifically at the *TNF- α* and *IL-1 β* promoters. Our results reveal a significant loss of H3K4me3 occupancy at the *TNF- α* and *IL-1 β* promoters in MLL1-depleted cells at 6 hours post-LPS induction compared to control cells pLKO1. This decrease in the activating histone mark did not correlate with the transcriptional output of *TNF- α* and *IL-1 β* . These results were incorporated in the revised manuscript results.

5. Figure 4c: The authors should show the relative mRNA levels of IL-1 β upon depletion of MLL1 and a ChIP-qPCR showing the loss of H3K4me3 at IL-1 β promoter.

Response:

We thank the reviewer for this suggestion. Following this recommendation, we have included both *IL-1 β* mRNA expression levels and H3K4me3 ChIP-qPCR data in the revised manuscript (Suppl. Fig. 7). Our results demonstrate that while MLL1 knockdown leads to a robust depletion of H3K4me3 at the *IL-1 β* promoter, this loss did not significantly impact its immediate transcriptional output under the conditions tested. These results have been added to the revised manuscript to further support our conclusions.

6. Figure 5/line 304-305: In the result section, the authors claimed that the results showed in figure 5 demonstrates that H3K4me3 deposition is delayed compared to transcription initiation, as it was shown for inducible genes. However, the experiment only shows that H3K4me3 is retained at MYC promoter after transcriptional shutdown.

Response:

We thank the reviewer for this observation. The text in the Results section has been revised to more accurately reflect the data presented in Figure 5. The statement has been modified to

indicate that, for the constitutively expressed MYC gene, H3K4me3 is retained at the promoter following transcriptional shutdown. This highlight of the temporal uncoupling between transcription and H3K4me3 levels ensures that the interpretation is fully supported by the experimental evidence provided.

7. The authors should refrain from using SEM in their figures, and show only SD instead. This allows a better estimation of the variance

Response:

We thank the reviewer for this comment. We have revised all figures to present data as mean \pm SD instead of SEM, allowing a more accurate representation of data variability

Minor comments:

• In figure 2 and S2, it would be clearer for the readers if the name of the genes corresponding to the data were added in the title of the figure or in the figure directly.

Response:

We thank the reviewer for this helpful suggestion. We have updated the titles of Figures 2 and S2 to clearly indicate the corresponding gene names, improving clarity and readability for the readers.

• In figure 3, the y-axis scales differ between the Fig3c and 3e (ActD and TPCA-1 input panels).

Response:

We thank the reviewer for pointing this out. We have now standardized the y-axis scales in Figures 3c and 3f (ActD and TPCA-1 input panels) to ensure consistent comparison and improve clarity.

• In the legend of figure S5., it is said that the blots correspond to data shown in figure 5a, it actually corresponds to figure 4a.

Response:

We thank the reviewer for noting this error. The legend of Figure S5 has been corrected to indicate that the immunoblots correspond to the data shown in Figure 4a, not Figure 5a.

Reviewer #2 (Comments to the Authors (Required)):

In this paper, the authors re-visit the role of histone H3K4me3 in transcriptional regulation. This is known to be deposited at active gene promoters and hence often implicated in the transcriptional activation process. However, there is a growing body of evidence that this mark may have a post-transcriptional role instead. Here, further evidence is gathered to support the latter hypothesis, by demonstrating that H3K4me3 deposition occurs later than transcriptional activation. This correlative observation is interesting, but the authors do not provide any data about what the role of this delayed deposition is. Overall, the study therefore adds further data to support a potential post-transcriptional role in gene regulation (or potentially no role and just a marker of previous transcription). It does not however provide a new paradigm as this observation has been made in other systems (as acknowledged in the discussion), and the authors should make this clearer in the abstract where they use the term "redefine" which would be better replaced with something like "confirm".

We sincerely thank the reviewer for careful reading of our manuscript and for recognizing the value of our data in the context of emerging role of the well-studied histone mark H3K4me3 in transcriptional regulation. We appreciate the acknowledgment that our observations are interesting and provide evidence supporting the emerging hypothesis that H3K4me3 deposition can occur subsequent to transcriptional activation in human cells. We agree that our study contributes to a growing body of literature suggesting that H3K4me3 may serve roles beyond initial activation, potentially acting as a post-transcriptional regulator or a molecular record of recent activity. Our primary objective was to characterize this temporal relationship within our specific system, and we value the reviewer's guidance in ensuring our findings are framed accurately within the broader field. In direct response to the reviewer's helpful suggestion, we have revised the Abstract and the manuscript to better reflect the scope and context of our results. We have also updated the discussion regarding the potential functional implications of this timing, openly acknowledging that while our data establishes a clear temporal sequence, the precise downstream consequences of this modification remain an important area for future investigation. We believe these revisions have significantly improved the clarity, balance, and scholarly impact of our manuscript.

*There are a number of specific issues that should be addressed:
(1) In Fig.1C (and SIC), there is a slight rise in H3K4me1 that precedes transcriptional*

activation. This should be commented on more as might be important. It might be that this mark matures later into H3K4me3? The latter possibility could also be added to the discussion.

Response:

We thank the reviewer for this insightful observation regarding the temporal dynamics of histone methylation. We agree that the modest increase in H3K4me1 preceding transcriptional activation (Fig. 1C, S1C) is potentially significant and likely reflects a "priming" or "poised" chromatin state upon induction. Following the reviewer's suggestion, we have expanded our Results and Discussion sections to explicitly address this early H3K4me1 signal. Furthermore, we have added a section to the Discussion regarding the possibility of a stepwise maturation from H3K4me1 to H3K4me3 as transcription becomes established. This interpretation aligns the observed temporal patterns with existing models of histone mark transition at active promoters.

(2) Fig.3d does not address transcriptional dependency as stated in the text. It assesses the role of NFkB. In that context, the role of NFkB might be to recruit MLL and this effect might itself be dependent or independent of transcriptional activity. The rationale therefore needs altering.

Response:

We thank the reviewer for this critique regarding the mechanistic role of NF- κ B. The rationale has been refined to clarify the distinction between NF- κ B recruitment and transcriptional dependency. While NF- κ B is necessary for the transcription, the kinetic data demonstrate a significant temporal dissociation (Fig. 1e and 2c): NF- κ B binding occurs rapidly aligning with the transcriptional peak, whereas it does not with the MLL1 and H3K4me3 enrichment.

If NF- κ B served as a direct physical scaffold for the immediate recruitment of MLL1, these events would be expected to coincide. The observed delay, combined with the loss of H3K4me3 following transcriptional blockade by Actinomycin D, indicates that NF- κ B occupancy alone is insufficient for mark deposition. Instead, the data suggest that H3K4me3 establishment is coupled to the active transcriptional process rather than being a recruitment event driven solely by NF- κ B. The Discussion sections have been updated to reflect this refined rationale.

(3) In Fig.4 what happens to H3K4me1/2 levels? Also, what about the levels of all H3K4me marks at enhancers surrounding these genes? What are their temporal dynamics?

Response:

We thank the reviewer for this important point. To examine the specificity of MLL1 knock down, we have now performed global Western blot analyses in shMLL1 cells for other H3K4me1/2, which confirmed that H3K4me1 levels remain stable while H3K4me2 slightly decreased, suggesting MLL1 is dispensable for the early establishment of 'priming' marks (H3K4me1/2) but is strictly required for the subsequent accumulation of H3K4me3. This observation is consistent with established models where enhancer-associated or priming marks are primarily deposited by MLL3/MLL4 complexes, whereas MLL1 is specifically recruited to regulate promoter-proximal trimethylation during active transcriptional induction (Herz et al., 2012; Calo and Wysocka, 2013). Similar 'maturation' dynamics have been reported for inducible inflammatory genes in macrophages (Kaikkonen et al., 2013).

Regarding the extension of ChIP-qPCR analysis to H3K4me1/2 and enhancer regions, several technical constraints were considered. As we noted in our manuscript (initial submission), MLL1 knockdown in THP-1 cells leads to a progressive decline in cell viability and increased cell death. Due to these limitations in obtaining sufficient yields of high-quality chromatin from viable cells, ChIP analysis was prioritized for H3K4me3 at the promoter region of *TNF- α* and *IL-1 β* . It showed significant reduction of H3K4me3 at the 6 h post-LPS induction in co comparison to the control pLKO.1 cells, concluding H3K4me3 is dispensable for their transcription. This data is shown as Fig. 4d-f and Suppl. Fig. 7b-c.

Regarding the enhancer regions around the *TNF- α* locus, it is characterized that *TNF- α* has a highly compact and modular architecture where essential regulatory elements and NF- κ B sites are clustered within the proximal promoter region (Falvo et al., 2010; Collart et al., 1990) and it is a small gene (2 kb). Given this integrated structure, our ChIP analysis focused on the 500bp promoter region, as it encompasses the primary regulatory hub for induction.

References

- Herz, H.-M. et al. *Genes & Development* 26, 2604–2620 (2012).
Calo, E. & Wysocka, J. *Molecular Cell* 49, 825–837 (2013).
Kaikkonen, M. U. et al. *Cell* 152, 131–145 (2013).

(4) I do not agree with the interpretation of Fig.5. The authors provide one explanation. However, equally plausible is that H3K4me3 turnover is slow and hence why it remains longer

than transcriptional activity. Is this not direct evidence of delayed deposition. I do not think this figure adds much to the paper.

Response:

We thank the reviewer for raising this important point regarding the interpretation of Figure 5. It is correctly noted that the persistence of the H3K4me3 signal relative to transcriptional output could reflect a slow turnover of the modification in addition to, or instead of, delayed deposition. We agree that the current data do not definitively distinguish between these two kinetic possibilities.

The primary objective of Figure 5 was to highlight the temporal separation between transcriptional activity and H3K4me3 occupancy. In this context, the data support the conclusion that H3K4me3 levels do not strictly mirror transcriptional kinetics in real-time. To avoid overinterpretation, the manuscript has been revised to clarify that while delayed deposition represents one plausible explanation, the potential contribution of slow H3K4me3 turnover must also be considered. This revision ensures a more precise interpretation of the results, highlighting that H3K4me3 occupancy reflects a distinct phase of the transcriptional cycle that does not strictly coincide with mRNA synthesis.

(5) The authors state that "binding" of MLL1 does not change in Supp Fig.S3c. However, the % input is very low, so this might represent no binding at all. This needs a negative control genomic region adding, to establish the basal level of antibody reactivity.

Response:

We thank the reviewer for this critique regarding the interpretation of the MLL1 ChIP signal. To address the concern that the low % input in Supplementary Fig. S3c might reflect background noise rather than specific binding, we have now included a negative control genomic region *MYOD* to establish the basal level of antibody reactivity. As *MyoD* is a lineage-specific gene silenced in myeloid cells, it provides a definitive baseline for non-specific antibody binding. Results demonstrate that MLL1 occupancy at the target promoters is significantly and consistently above this *MyoD* background. Furthermore, the *GAPDH* promoter was utilized as a constitutive positive control; the clear enrichment at this locus confirms the antibody's specificity for active promoter regions and validates the experimental protocol.

(6) Overall, again acknowledged by the authors, this study is limited to a couple of examples which may be atypical. It would have been better to perform RNA-seq and ChIP-seq for H3K4me3 across an activation timecourse to provide more definitive results.

Response

We agree with the reviewer that genome-wide RNA-seq and H3K4me3 ChIP-seq time-course analyses would further generalize our conclusions. However, the primary objective of the current study was to establish a high-resolution mechanistic foundation by focusing on the precise kinetics of representative model loci. While large-scale genomic studies are beyond the current scope, we recognize the significant value of this suggestion and have expanded the Discussion to highlight genome-wide characterization as a critical next step for future investigation. We believe this addition provides a more forward-looking for better understanding of the role of H3K4me3 in transcription.

*Reviewer #3 (Comments to the Authors (Required)):
Summary*

Walvekar et al. used two inducible human genes (TNF- α and IL-1 β) to examine the timing of H3K4me3 enrichment relative to transcription. They conducted time-course profiling of transcriptional activity and histone modifications at one location downstream of the transcription start sites of TNF- α and IL-1 β . This setup shows that H3K4me2 and H3K4me3 deposition and MLL1 occupancy do not align with the activation of these genes. Instead, MLL1 occupancy and H3K4me2/3 enrichment are delayed, peaking about 6 hours after gene induction. In contrast, p300 and histone acetylation (H3K27ac and H4K8ac) follow the same pattern as mRNA transcription, peaking at 2 hours after LPS stimulation. Finally, inhibiting RNA synthesis with Actinomycin D, along with time-course profiling of transcription activity and histone modifications downstream of MYC's transcription start site, shows that H3K4me3 remains present roughly 2 hours after transcriptional inhibition. This temporal separation between transcription and deposition of H3K4me3 supports the conclusion that H3K4me3 functions as a downstream marker of active transcription rather than a prerequisite for its initiation and therefore represents a "post-transcriptional histone mark".

In summary, the authors provide another example of a conclusion that the field has already reached (1,2). In fact, the authors discuss relevant literature in their discussion. Therefore, the central claim that H3K4me3 follows transcription and depends on prior RNA synthesis, serving

as a post-transcriptional histone mark rather than a driver of expression, is not new. However, there is some value in demonstrating this in a temporal manner at inducible human genes.

We appreciate the reviewer's summary and the recognition of the value in our temporal analysis. While the relationship between transcription and H3K4me3 has been explored, the significance of our study lies in the direct evidence of temporal uncoupling observed at the investigated loci.

Specifically, the data show that H3K4me3 levels peak at 6 hours, significantly lagging behind the peak of mRNA synthesis at 2 hours for both *TNF- α* and *IL-1 β* . Furthermore, the persistence of H3K4me3 at the MYC promoter for at least four hours after transcriptional shutdown demonstrates that the mark does not track with mRNA synthesis in real-time. These findings provide clear evidence that, at these promoters, H3K4me3 is a downstream consequence of the transcriptional process rather than a requirement for its initiation.

Comments

1. The main claim of this manuscript is that H3K4me3 follows transcription and depends on prior RNA synthesis, serving as a post-transcriptional histone mark rather than a driver of expression. In my opinion, the field has already concluded this from studies over the last two decades. H3K4 methylation has been demonstrated to be dispensable for transcriptional activation from yeast to mouse (3-9). In the context of MLL2, H3K4me3 does not activate gene expression but instead prevents a default repressed state induced by de novo DNA methyltransferases and PRC2 activity (10). Thus, H3K4 methylation is not required for transcriptional activation and instead regulates promoter-proximal pausing through a mechanism involving the recruitment of the Integrator complex, which is essential for the eviction of paused RNAPII and transcriptional elongation (11).

Previous evidence for the deposition of H3K4me3 as a result of transcription comes from time-resolved experiments in yeast, Drosophila, and mouse. Research on the oscillating expression of over half of budding yeast genes during the metabolic cycle shows that, for most of these genes, H3K4me3 peaks after the maximum of their RNA transcripts (12). During meiosis in yeast, a delay in H3K4me3 compared to induced RNA is also observed (13). In zebrafish, bulk H3K4me3 is absent before the initiation of transcription during the maternal-zygotic transition (14). Finally, for genes with expression changes during diurnal cycles in the mouse liver,

H3K4me3 generally peaked 1.3 hours after RNA polymerase II occupancy (15).

Mellor et al. have discussed the non-instructive role for H3K4me3 in transcription (1). I directly quote: "Another possibility is that, although H3K4me3 and transcription correlate strongly, these are actually independent events. A simple explanation is that an NDR over DNA with a suitable base composition, such as a CpG island in mammals, recruits SET1A/B leading to H3K4me3. Independently, an NDR might also be sufficient to recruit RNA polymerase II, either supported by DNA-bound transcription factors or not, leading to transcription, hence the apparent link between H3K4me3 and transcription. However, as the non-perfect correlation between H3K4me3 and transcription suggests, it is possible to get H3K4me3 without transcription and vice versa. Evidence for this comes from mESCs: at engineered CpG islands with no nearby promoters, CFP1-dependent H3K4me3 can be deposited when surrounding transcription levels, assessed using RNA polymerase II ChIP, are very low or absent and this H3K4me3 does not, in turn, lead to transcription" (16).

Lastly, the authors also acknowledge that the dynamics and regulatory roles of H3K4me3 need to be studied across a broader range of active promoters, ideally at genome-wide resolution. Based on the above comments, the central claim of the manuscript is not new, but there is some value in demonstrating it again, in a temporal manner, using two inducible human genes.

Response:

We thank the reviewer for this comprehensive and scholarly overview of the literature regarding the non-instructive role of H3K4me3. We acknowledge the landmark studies cited (3–16) which established that H3K4 trimethylation is often dispensable for transcriptional initiation and can lag behind mRNA synthesis in various model organisms.

However, the significance of our study lies in the magnitude and clarity of the temporal uncoupling observed specifically in the context of acute human inflammatory induction. While diurnal cycles in mouse liver show a modest 1.3-hour lag but some overlap with transcriptional peak (15), our data demonstrate a substantial 4-hour delay between the peak of mRNA synthesis (2 hours) and the peak of H3K4me3 occupancy (6 hours) for *TNF- α* and *IL-1 β* . This extended kinetic window provides a unique opportunity to distinguish transcription (and acetylation and Pol II) and the significantly delayed H3K4me3 deposition on chromatin landscape.

Furthermore, our finding that H3K4me3 remains stable at the *MYC* promoter long after transcriptional shutdown adds a crucial dimension to the "post-transcriptional" model: it suggests that in human cells, the mark functions not just as a byproduct of the current round of transcription, but as a relatively stable epigenetic signature of prior activity. We believe that by providing these high-resolution kinetics in a human system, our work offers a definitive physiological example of uncoupling that complements the broader conceptual framework established by previous studies in other models.

2. I recommend revising the title, "Human Inducible Gene Model Identifies H3K4me3 as a Post-Transcriptional Histone Mark," to a more suitable alternative. A new title that better reflects the scope and findings could do justice to the study without overselling the claims.

Response:

We thank the reviewer for the suggestion to revise the title to better reflect the scope of our findings. We have updated the title to emphasize the temporal dynamics of the process and the specific recruitment and deposition events we observed at human inducible loci. We believe this title accurately represents our data.

“Temporal analysis of inducible human genes reveals post-transcriptional H3K4me3 deposition”

3. The authors claim to have performed "high-resolution" analyses of transcriptional activity and histone modifications in human cells. However, their analyses focus on two loci approximately 500 bp upstream of the corresponding transcription start sites (TSS). What is the reason for choosing these promoter locations? H3K4me3 shows greater enrichment at the +1 nucleosome just downstream of the nucleosome-depleted region (NDR). The authors should include at least a second region downstream of the TSS in their histone modification profiling studies.

Response:

We thank the reviewer for this insightful comment regarding our spatial resolution. The selection of the –500 bp to +500 bp region was selected by the high-resolution H3K4me3 ChIP-seq profiles from human monocytes available via the UCSC Genome Browser (ENCODE Epigenome Project).

Based on this empirical data on the H3K4me3 enrichment profiles in CD14+ monocytes we have designed our primers that show overlap with the H3K4me3 peaks at the *TNF- α* and *IL-1 β* promoters. The UCSC gene maps of the analysed loci and regions of analysis with H3K4me3 enrichment are now clearly shown as a Suppl. Figure 1. Furthermore, because our ChIP assays utilize sonicated chromatin with an average fragment size of ~500 bp, the enrichment measured at these loci inherently captures the signal from the nucleosome-depleted region (NDR) and the adjacent +1 nucleosome. This ensures that our analysis accurately reflects the chromatin environment most critical for transcription initiation and early elongation.

This strategy is further supported by established genome-wide profiles showing that H3K4me3 enrichment typically spans -1 kb to +1 kb around the TSS, with core enrichment concentrated in the promoter-proximal region (Refs 1, 2).

Nevertheless, to fully address the reviewer's suggestion and provide even more comprehensive profiling, we have included new ChIP data for an additional downstream region for both *TNF- α* and *IL-1 β* that as well overlap with the H3K4me3 peak. These new results demonstrate temporal dynamics entirely consistent with our initial findings and have been incorporated into the revised manuscript (Suppl. Fig. 3).

1. Yu H, Lesch BJ. Functional Roles of H3K4 Methylation in Transcriptional Regulation. *Mol Cell Biol.* 2024;44(11):505-515. doi: 10.1080/10985549.2024.2388254. Epub 2024 Aug 18. PMID: 39155435; PMCID: PMC11529435.

2. Benayoun, B. A. et al. H3K4me3 breadth is linked to cell identity and transcriptional consistency. *Cell* **158**, 673–688 (2014).

4. In Figure 4, the authors generated stable THP-1 monocytic cell lines expressing two shRNAs targeting human MLL1 using the pLKO.1 lentiviral vector system. Western blot analysis confirmed the reduction of MLL1 by ~50%, which is partially accompanied by a global decrease in H3K4me3 levels compared to control cells transduced with the empty vector (Fig. 4a-b). The three Western blots clearly show differences in the knockdown efficiency of MLL1 (~50%-75%) and, consequently, global H3K4me3 levels (~30%-75% reduction). It is unclear to the reader whether the three western blots correspond to six independent stable THP-1 lines

and which cell lines were used for densitometric quantification in Figure 4b. Is the data displayed as the mean from three independent stable THP-1 lines?

Response:

We thank the reviewer for pointing out this ambiguity and agree that clarification is needed. The three Western blots shown in Fig. 4a represent three independent experiments (n = 3) performed using the same stable THP-1 cell lines expressing shRNAs targeting MLL1 or the empty vector control. The variability in knockdown efficiency observed across the blots reflects experimental variability between independent experiments rather than differences between distinct stable cell lines.

Densitometric quantification shown in Fig. 4b was performed using the data from these three independent experiments, and the values are presented as mean \pm SD. We have now clarified this explicitly in the figure legend to avoid confusion.

Importantly, despite some variability in the extent of MLL1 knockdown and global H3K4me3 reduction, the overall trend is consistent across experiments, supporting the robustness of the observed effect.

In Figure 4c, a quantitative RT-PCR analysis of TNF- α mRNA expression in control and shMLL1 cells following LPS stimulation (2 h and 6 h) is shown. Have the authors measured the effect of MLL1 knockdown on the H3K4me3 levels at the TNF- α promoter following LPS stimulation? I am missing this important experiment here to conclude that H3K4me3 depletion does not impair LPS-induced TNF- α mRNA expression.

Response:

To address the reviewer's comment, we have expanded Figure 4 to include CHIP analysis to quantify H3K4me3 enrichment specifically at the *TNF- α* and *IL-1 β* promoters. Our results reveal a significant loss of H3K4me3 occupancy at the *TNF- α* and *IL-1 β* promoters in MLL1-depleted cells at 6 hours post-LPS induction compared to control cells PLKO1. This decrease in the activating histone mark did not correlate with the transcriptional output of *TNF- α* and *IL-1 β* . We have updated the Results accordingly. Inclusion of this experiment strengthens the link between chromatin changes and transcriptional output in our system.

5. The antibodies used in this study appear to have lower specificity compared to other commercial antibodies (<http://www.histoneantibodies.com/FinalArrayData/H3K4me3/>). The authors could use another H3K4me3 with higher specificity to confirm their results on delayed H3K4me3 deposition after LPS stimulation, ideally at two promoter locations (*{plus minus}* 500 bp upstream and downstream of the TSS) of the *TNF- α* and *IL-1 β* genes.

Response:

We thank the reviewer for this thoughtful comment regarding antibody specificity. We have addressed these concerns as follows:

First, the H3K4me3 antibody used in this study (Cell Signaling Technology, CST) is a well-validated gold standard in the field. It has been utilized extensively in high-impact CHIP-seq studies (1, 2) and was a primary antibody for the UCSC/ENCODE Epigenome Project. The enrichment patterns we observed—specifically the promoter-proximal peaks centered around transcription start sites (TSS)—are entirely consistent with these established genomic distributions.

Second, we would like to emphasize that the central conclusion of our study—the delayed deposition of H3K4me3—is based on relative temporal changes measured using the same antibody across all conditions (basal vs. LPS-stimulated). Because the antibody and experimental parameters remained constant, any inherent differences in absolute specificity compared to other commercial lots would not alter the observed kinetic shift.

Finally, the reproducibility of this delayed accumulation across multiple independent inflammatory loci (e.g., *TNF- α* and *IL-1 β*) strongly suggests a conserved biological mechanism. The fact that H3K4me3 enrichment correlates with, rather than precedes, transcriptional initiation supports our conclusion that this mark is a consequence, rather than a prerequisite, of the transcriptional process in this context.

Ref:

1. Cheng Z, Fang Z, Yue K, Guo Y, Huang L, Zhang Y. Epigenetic regulation of TIPE3 in nasopharyngeal carcinoma and its impact on the hedgehog signaling pathway. *Cell Mol Life*

Sci. 2025 Nov 13;82(1):394. doi: 10.1007/s00018-025-05924-1. PMID: 41231288; PMCID: PMC12615883.

2. Joun GL et. all Histone methyltransferase PRDM9 promotes survival of drug-tolerant persister cells in glioblastoma. Nat Commun. 2025 Dec 15;16(1):10905. doi: 10.1038/s41467-025-65888-5. PMID: 41397959; PMCID: PMC12705669.

6. Hypermethylated H3K4 within the mRNA coding region of yeast genes persists for a considerable time after transcription is shut down and Set1 dissociates from chromatin (17), indicating that H3K4 methylation acts as a molecular memory of recent transcriptional activity, as the authors suggest. The authors could have further investigated the potential role of H3K4 methylation as a molecular memory, but they do not. Can the authors provide additional experimental evidence that H3K4 methylation functions as a molecular memory in inducible human genes?

Response:

We thank the reviewer for this insightful comment. We agree that H3K4 methylation has been proposed to function as a molecular memory of recent transcriptional activity, primarily based on studies in yeast. Our data provide temporal evidence consistent with this model in human cells that H3K4me3 is deposited as a consequence of transcriptional initiation and depending on the prior transcriptional event. Based on this we speculated the same as a possibility that H3K4me3 may reflect recent transcriptional history/memory. However, we would like to clarify that we do not claim that H3K4 methylation functions as a molecular memory in inducible human genes. Directly testing this would require additional functional perturbation and re-induction experiments beyond the scope of the present study, which we noted it as an important direction for future work.

1. Howe FS, Fischl H, Murray SC, Mellor J. Is H3K4me3 instructive for transcription activation? BioEssays. 2017 Jan;39(1):1-12.

2. Morgan MAJ, Shilatifard A. Reevaluating the roles of histone-modifying enzymes and their associated chromatin modifications in transcriptional regulation. Nat Genet. 2020 Dec;52(12):1271-81.

3. Miller T, Krogan NJ, Dover J, Erdjument-Bromage H, Tempst P, Johnston M, et al. COMPASS: A complex of proteins associated with a trithorax-related SET domain protein.

- Proc Natl Acad Sci.* 2001 Nov 6;98(23):12902-7.
4. Nislow C, Ray E, Pillus L. *SET1*, A Yeast Member of the Trithorax Family, Functions in Transcriptional Silencing and Diverse Cellular Processes. *Mol Biol Cell.* 1997 Dec;8(12):2421-36.
 5. Roguev A. The *Saccharomyces cerevisiae* Set1 complex includes an Ash2 homologue and methylates histone 3 lysine 4. *EMBO J.* 2001 Dec 17;20(24):7137-48.
 6. Briggs SD, Bryk M, Strahl BD, Cheung WL, Davie JK, Dent SYR, et al. Histone H3 lysine 4 methylation is mediated by Set1 and required for cell growth and rDNA silencing in *Saccharomyces cerevisiae*. *Genes Dev.* 2001 Dec 15;15(24):3286-95.
 7. Hödl M, Basler K. Transcription in the Absence of Histone H3.2 and H3K4 Methylation. *Curr Biol.* 2012 Dec;22(23):2253-7.
 8. Denissov S, Hofemeister H, Marks H, Kranz A, Ciotta G, Singh S, et al. Mll2 is required for H3K4 trimethylation on bivalent promoters in embryonic stem cells, whereas Mll1 is redundant. *Development.* 2014 Feb 1;141(3):526-37.
 9. Hu D, Garruss AS, Gao X, Morgan MA, Cook M, Smith ER, et al. The Mll2 branch of the COMPASS family regulates bivalent promoters in mouse embryonic stem cells. *Nat Struct Mol Biol.* 2013 Sept;20(9):1093-7.
 10. Douillet D, Sze CC, Ryan C, Piunti A, Shah AP, Ugarenko M, et al. Uncoupling histone H3K4 trimethylation from developmental gene expression via an equilibrium of COMPASS, Polycomb and DNA methylation. *Nat Genet.* 2020 June;52(6):615-25.
 11. Wang H, Fan Z, Shliaha PV, Miele M, Hendrickson RC, Jiang X, et al. H3K4me3 regulates RNA polymerase II promoter-proximal pause-release. *Nature.* 2023 Mar 9;615(7951):339-48.
 12. Kuang Z, Cai L, Zhang X, Ji H, Tu BP, Boeke JD. High-temporal-resolution view of transcription and chromatin states across distinct metabolic states in budding yeast. *Nat Struct Mol Biol.* 2014 Oct;21(10):854-63.
 13. Borde V, Robine N, Lin W, Bonfils S, Géli V, Nicolas A. Histone H3 lysine 4 trimethylation marks meiotic recombination initiation sites. *EMBO J.* 2009 Jan 21;28(2):99-111.
 14. Vastenhouw NL, Zhang Y, Woods IG, Imam F, Regev A, Liu XS, et al. Chromatin signature of embryonic pluripotency is established during genome activation. *Nature.* 2010 Apr 8;464(7290):922-6.
 15. Le Martelot G, Canella D, Symul L, Migliavacca E, Gilardi F, Liechti R, et al. Genome-Wide RNA Polymerase II Profiles and RNA Accumulation Reveal Kinetics of Transcription

and Associated Epigenetic Changes During Diurnal Cycles. Kramer A, editor. PLoS Biol. 2012 Nov 27;10(11):e1001442.

16. Thomson JP, Skene PJ, Selfridge J, Clouaire T, Guy J, Webb S, et al. CpG islands influence chromatin structure via the CpG-binding protein Cfp1. Nature. 2010 Apr;464(7291):1082-6.

17. Ng HH, Robert F, Young RA, Struhl K. Targeted Recruitment of Set1 Histone Methylase by Elongating Pol II Provides a Localized Mark and Memory of Recent Transcriptional Activity. Mol Cell. 2003 Mar;11(3):709-19.

April 7, 2026

RE: Life Science Alliance Manuscript #LSA-2025-03511-TR

Dr. Sabarinadh Chilaka
Indian Institute of Chemical Technology
Applied Biology
Tarnaka
Hyderabad, Telangana 500007
India

Dear Dr. Chilaka,

Thank you for submitting your revised manuscript entitled "Temporal analysis of inducible human genes reveals post-transcriptional H3K4me3 deposition". It has now been evaluated by all the original reviewers whose comments are appended below. As you will read, Reviewer 1 and 2 acknowledge that your revised work has addressed their previous concerns. Reviewer 3 highlights overstatement and over-generalisation of multiple conclusions in points 1-5. We agree with this reviewer that you must address these concerns with appropriate textual changes and tone down conclusions throughout the manuscript text.

We would be happy to publish your paper in Life Science Alliance pending resolution of the above points and final revisions necessary to meet our formatting guidelines. We request you to submit a revised manuscript document with all these changes highlighted along with a point-by-point response to the reviewer's comments.

MANUSCRIPT ORGANIZATION AND FORMATTING:

To avoid unnecessary delays in the acceptance and publication of your paper, please read the following information carefully. Full guidelines are available on our Instructions for Authors page, <https://www.life-science-alliance.org/authors>

-In Figure 4A, the GAPDH row in the top panel appears to be duplicated. Please verify and remove the duplicated row. Otherwise, please explain in the legends.

-We recommend that you rename Figure S8 as a source data file instead of a supplementary figure. Accordingly please change the callout on page 19.

-LSA does not permit citation of "data not shown," "manuscript in preparation," "manuscript submitted," etc., in any section of the manuscript. Please provide evidence to back the statement on page 10 about MLL1 knockdown cells exhibiting a progressive decline in viability and proliferation. If you cannot provide the evidence to back the claim, please remove it.

-In the methods section, please describe how western blots were quantified to determine %input.

-Please explain the axis title %input at least in the legend for Figure 1.

-Please include information on antibody concentration/amount used in Suppl Table 3.

-Please upload your main manuscript text as an editable doc file.

-Please upload your main and supplementary figures as single files.

-Please add the X and Bluesky handles of your host institute/organization, as well as your own, and/or one of the authors, in our system

-Please upload a clean manuscript file without the highlighted text. The version with highlighted changes you can upload with the file designation "Related Manuscript File."

-Please add your main and supplementary figure legends to the main manuscript text after the references section.

-Abstract should be a single paragraph not exceeding 175 words. Please update the system as well to match the manuscript file.

-Please rename "Competing interests" to "Conflict of Interest"

-Please add callouts for Figures 1H; 2F; 3C, F; 4E; 5D; S1A-C; S4E; S6C, F and S7C to your main manuscript text.

-Please be sure that the authorship listing and order is correct

We welcome submissions of potential cover images for the issue of LSA in which your work would appear. If you have high quality images associated with this work, please feel free to email these, with a caption, to the journal office.

LSA encourages authors to provide a 30-60 second video where the study is briefly explained. We will use these videos on social media to promote the published paper and the presenting author (for examples, see <https://docs.google.com/document/d/1-UWCfbE4pGcDdcgzcmiuJl2XMBJnxKYeqRvLLrLSo8s/edit?usp=sharing>). Corresponding or first-authors are welcome to submit the video. Please submit only one video per manuscript. The video can be emailed to contact@life-science-alliance.org

FINAL FILES:

The following items are required for acceptance.

The license to publish form must be signed before your manuscript can be sent to production. A link to the license to publish form will be available to the corresponding author only. Please take a moment to check your funder requirements.

Thank you for your attention to these final processing requirements. Please revise and format the manuscript and upload materials as soon as you are able.

Thank you for this interesting contribution to the literature. We look forward to publishing your paper in Life Science Alliance.

Sincerely,

Sarita Hebbar, PhD
Scientific Editor
Life Science Alliance
<http://www.lsjournal.org>

Reviewer #1 (Comments to the Authors (Required)):

The authors have addressed all requests/comments.

Reviewer #2 (Comments to the Authors (Required)):

The authors have provided a good response to my comments and the contents of the paper are now reflective of the conclusions. caveats and future experimentation possibilities are also increased. Extra control data is added. Overall this provides a useful contribution that supports existing findings in this area but with a new system being studied.

Reviewer #3 (Comments to the Authors (Required)):

The main claim of this manuscript is that H3K4me3 follows transcription and depends on prior RNA synthesis, serving as a post-transcriptional histone mark rather than a driver of expression. I stand by my previous assessment that the central conclusion is consistent with a substantial body of prior work demonstrating that H3K4 methylation is dispensable for transcriptional initiation and can follow transcription. Therefore, the conceptual advance is limited. However, the manuscript provides a temporally resolved analysis of human inducible genes, which represents a more incremental but still useful contribution.

Comments

1. The authors used two inducible human genes (TNF- α and IL-1 β) to examine the timing of H3K4me3 enrichment relative to transcription. I still find some of the statements and conclusions too general. The manuscript repeatedly extrapolates from two inducible inflammatory loci to general properties of H3K4me3 across the human genome. While the authors appropriately acknowledge this limitation in the Discussion, the title, abstract, and several statements remain overly general and should be revised to reflect the locus-specific nature of the data. I recommend revising the new title, "Temporal analysis of inducible human genes reveals post-transcriptional H3K4me3 deposition" to a more suitable alternative like: "Temporal analysis of two inducible human genes reveals post-transcriptional H3K4me3 deposition". Could the authors provide data for more inducible human genes? TNF- α and IL-1 β could be outliers.
2. In Figure 4 and Supplementary Figure 7, the authors conclude that H3K4me3 is dispensable for the initiation of TNF- α and IL-1 β transcription. However, the conclusion that H3K4me3 is dispensable for transcriptional initiation is not fully supported by the data. MLL1 depletion results in only partial reduction of promoter H3K4me3 at early time points (~2 h), when transcription is assessed by mRNA levels. Therefore, the experiment does not test transcriptional initiation in the absence of H3K4me3, but rather under conditions of residual mark. A more complete depletion or orthogonal perturbation would be required to support this claim.
3. The authors claim to have performed "high-resolution" analyses of transcriptional activity and histone modifications in human cells. The temporal sampling (hour-scale intervals) and locus-restricted ChIP-qPCR measurements do not meet the standards typically associated with "high-resolution" chromatin or transcriptional analyses. Therefore, I recommend avoiding the term "high-resolution".
4. While the temporal uncoupling between transcription and H3K4me3 deposition is clearly demonstrated, the manuscript does not provide mechanistic insight into the functional consequences of this delayed deposition. As such, the study remains descriptive and does not resolve whether H3K4me3 plays a post-transcriptional regulatory role or serves as a passive mark of prior transcription.
5. The persistence of H3K4me3 following transcriptional inhibition (Figure 5) does not distinguish between delayed deposition and slow turnover of the modification. Therefore, this experiment does not provide direct evidence supporting a post-transcriptional deposition mechanism.

In summary, the manuscript provides a clear temporal analysis of H3K4me3 dynamics at two inducible human loci and supports the view that H3K4me3 follows transcription. However, the conceptual advance relative to prior literature is limited, the conclusions are overgeneralized beyond the data, and key mechanistic claims (e.g., dispensability and post-transcriptional function) are not fully supported by the presented experiments.

Authors sincerely thank the Editor-in-Chief, handling editor, and reviewers for their constructive comments and valuable suggestions to enhance our manuscript. We are also grateful for the encouraging feedback we received.

As per the LSA journal norms, we have prepared a point-by-point response to each of the reviewer's comments.

- Reviewer comments are in '*blue and italics*' and author responses are in black and plain text
- **Yellow highlights** in the manuscript represent modified text or added text as per the review comments in the revised manuscript

Current title of the manuscript: **Temporal analysis of two inducible human genes reveals post-transcriptional H3K4me3 deposition**

Submission ID: **LSA-2025-03511-T R2**

MANUSCRIPT ORGANIZATION AND FORMATTING:

Response:

In Figure 4A, the GAPDH row in the top panel appears to be duplicated. Please verify and remove the duplicated row. Otherwise, please explain in the legends.

We thank the editor for pointing this out. The duplication of the GAPDH row in the top panel of Figure 4A was mistakenly duplicated during figure preparation. The duplicated row has now been removed, and the corrected figure has been updated in the revised manuscript.

-We recommend that you rename Figure S8 as a source data file instead of a supplementary figure. Accordingly please change the callout on page 19.

We thank the Editor for this suggestion. Figure S8 has now been renamed as a source data file, and the corresponding callout on page 19 has been updated accordingly in the revised manuscript.

-LSA does not permit citation of "data not shown," "manuscript in preparation," "manuscript submitted," etc., in any section of the manuscript. Please provide evidence to back the statement on page 10 about MLL1 knockdown cells exhibiting a progressive decline in viability and proliferation. If you cannot provide the evidence to back the claim, please remove it.

The phrase "data not shown" has been removed from the statement on page 10, and the text has been revised accordingly to comply with the journal guidelines.

-In the methods section, please describe how western blots were quantified to determine %input. -Please explain the axis title %input at least in the legend for Figure 1.

The description of how western blots were quantified to determine fold change values has now been included in the Methods section, and the % input axis has been clearly explained in the legend for Figure 1 in the revised manuscript.

-Please include information on antibody concentration/amount used in Suppl Table 3.

The information on antibody concentration/amount used has now been included in Supplementary Table 3 in the revised manuscript.

-Please upload your main manuscript text as an editable doc file.

The main manuscript text has been uploaded as an editable document file.

-Please upload your main and supplementary figures as single files.

The main and supplementary figures has been uploaded as single files accordingly.

-Please add the X and Bluesky handles of your host institute/organization, as well as your own, and/or one of the authors, in our system

The X and Bluesky handles of the host institute/organization and the authors will be added in the system accordingly.

-Please upload a clean manuscript file without the highlighted text. The version with highlighted changes you can upload with the file designation "Related Manuscript File."

A clean version of the manuscript without highlighted text has been uploaded, and the version with highlighted changes has been uploaded as a "Related Manuscript File."

-Please add your main and supplementary figure legends to the main manuscript text after the references section.

The main and supplementary figure legends have now been added to the main manuscript text after the references section in the revised version.

-Abstract should be a single paragraph not exceeding 175 words. Please update the system as well to match the manuscript file.

The abstract has been revised to a single paragraph not exceeding 175 words, and the system entry has been updated accordingly to match the manuscript file.

-Please rename "Competing interests" to "Conflict of Interest"

"Competing interests" has been renamed to "Conflict of Interest" in the revised manuscript.

-Please add callouts for Figures 1H; 2F; 3C, F; 4E; 5D; S1A-C; S4E; S6C, F and S7C to your main manuscript text.

Callouts for Figures 1H, 2F, 3C, F, 4E, 5D, S1A–C, S4E, S6C, F, and S7C have now been added to the main manuscript text in the revised version.

-Please be sure that the authorship listing and order is correct

The authorship listing and order have been carefully checked and confirmed to be correct in the revised manuscript.

Response to Reviewers

Reviewer #1

The authors have addressed all requests/comments.

We thank the reviewer for the time and thoughtful feedback provided throughout the peer-review process. We are pleased that our revisions have addressed all requests and comments to the reviewer's satisfaction, and we believe their suggestions have significantly improved the clarity and quality of the manuscript. we appreciate the positive assessment of our work.

Reviewer #2

The authors have provided a good response to my comments and the contents of the paper are now reflective of the conclusions. caveats and future experimentation possibilities are also increased. Extra control data is added. Overall this provides a useful contribution that supports existing findings in this area but with a new system being studied.

We thank the reviewer for the thoughtful evaluation of our revised manuscript. We believe Reviewer's suggestions have significantly improved the manuscript, and we appreciate the positive assessment that this work provides a useful contribution to the field by extending existing findings to a new experimental system.

Reviewer #3

The main claim of this manuscript is that H3K4me3 follows transcription and depends on prior RNA synthesis, serving as a post-transcriptional histone mark rather than a driver of expression. I stand by my previous assessment that the central conclusion is consistent with a substantial body of prior work demonstrating that H3K4 methylation is dispensable for transcriptional initiation and can follow transcription. Therefore, the conceptual advance is limited. However, the manuscript provides a temporally resolved analysis of human inducible genes, which represents a more incremental but still useful contribution.

We thank the reviewer for the evaluation and for the thoughtful perspective on the conceptual positioning of our work. We appreciate the acknowledgment that our temporally resolved analysis of H3K4me3 using human inducible genes presents a useful contribution to the field. We agree with the reviewer's point that our findings align with the established view that H3K4 methylation is dispensable for transcriptional initiation; however, we believe that by precisely

defining the kinetic lag in a human system, our study provides a necessary empirical bridge for these existing models and for future investigations.

Comments

1. The authors used two inducible human genes (TNF- α and IL-1 β) to examine the timing of H3K4me3 enrichment relative to transcription. I still find some of the statements and conclusions too general. The manuscript repeatedly extrapolates from two inducible inflammatory loci to general properties of H3K4me3 across the human genome. While the authors appropriately acknowledge this limitation in the Discussion, the title, abstract, and several statements remain overly general and should be revised to reflect the locus-specific nature of the data. I recommend revising the new title, "Temporal analysis of inducible human genes reveals post-transcriptional H3K4me3 deposition" to a more suitable alternative like: "Temporal analysis of two inducible human genes reveals post-transcriptional H3K4me3 deposition". Could the authors provide data for more inducible human genes? TNF- α and IL-1 β could be outliers.

Response:

We thank the reviewer for this helpful suggestion. We agree that our original title was overly general relative to the scope of the data. In line with the reviewer's recommendation, we have revised the title to: "Temporal analysis of two inducible human genes reveals post-transcriptional H3K4me3 deposition." This revised title more accurately reflects the locus-specific nature of our study, which focuses on the inducible inflammatory genes *TNF- α* and *IL-1 β* . It also aligns with our central observation that H3K4me3 accumulation follows transcriptional activation in these systems. We have further revised the Abstract and Discussion to avoid overgeneralization and to clarify the scope and interpretation of our findings.

We acknowledge the reviewer's concern regarding the number of loci. However, our study was designed to prioritize temporal precision over genomic breadth. We specifically selected *TNF- α* and *IL-1 β* as transcriptional models for several strategic reasons that mitigate the risk of them being "outliers": **Synchronous "Off-On-Off" Kinetics:** As noted in our Discussion, these loci provide a precise, well-characterized system of rapid and transient induction. Their short mRNA half-lives (Adamik et al., 2013; Suzuki et al., 2000) allow us to capture the exact transition from initiation to the post-transcriptional phase, a resolution that is technically obscured in constitutively expressed or slower-responding genes.

Chromosomal Independence: These two genes are located on different chromosomes (Chr 6 and Chr 2). The fact that they exhibit identical H3K4me3 loading kinetics (lagging significantly behind Pol II recruitment and RNA synthesis) indicates a conserved regulatory mechanism rather than a site-specific chromosomal artifact.

Native Context: By studying these genes at their endogenous loci, we preserved the native chromatin architecture. This avoided the common artefacts associated with transgene-based systems, ensuring that the observed temporal uncoupling reflects physiological epigenetic dynamics.

By focusing on these gold-standard models, we were able to perform multiple functional perturbations (ActD, MLL1-KD, NF- κ B blockade) with a sampling frequency that provides a definitive kinetic framework, which would be technically diluted in a broader, less frequent genomic analysis. We have further improved the discussion on this point.

2. In Figure 4 and Supplementary Figure 7, the authors conclude that H3K4me3 is dispensable for the initiation of TNF- α and IL-1 β transcription. However, the conclusion that H3K4me3 is dispensable for transcriptional initiation is not fully supported by the data. MLL1 depletion results in only partial reduction of promoter H3K4me3 at early time points (~2 h), when transcription is assessed by mRNA levels. Therefore, the experiment does not test transcriptional initiation in the absence of H3K4me3, but rather under conditions of residual mark. A more complete depletion or orthogonal perturbation would be required to support this claim.

Response:

We thank the reviewer for this critical observation regarding the technical limitations of the MLL1 knockdown system. We agree that because a total loss of H3K4me3 was not achieved, the experimental conditions reflect a state of significant depletion rather than absolute absence. Consequently, we have revised our conclusions and adjusted our terminology in the revised manuscript to state that transcriptional activation occurs independently of maximal H3K4me3 enrichment, rather than claiming absolute dispensability. However, our findings in MLL1-depleted cells demonstrate a lack of quantitative coupling between H3K4me3 density and transcriptional output. While residual levels of the mark persist in the MLL1-depleted cells, the fact that a substantial reduction in H3K4me3 failed to impair the induction of *TNF- α* and *IL-1 β* suggests the mark is not a rate-limiting prerequisite for initiation. If H3K4me3 functioned as an inductive trigger, even partial depletion should dampen the transcriptional response.

Instead, achieving full mRNA induction despite significant loss of the mark supports a model where high-density H3K4me3 serves as a stable readout of sustained activity rather than a requirement for the inflammatory program to begin.

3. The authors claim to have performed "high-resolution" analyses of transcriptional activity and histone modifications in human cells. The temporal sampling (hour-scale intervals) and locus-restricted ChIP-qPCR measurements do not meet the standards typically associated with "high-resolution" chromatin or transcriptional analyses. Therefore, I recommend avoiding the term "high-resolution".

Response:

We agree with the reviewer that the term "high-resolution" can be subjective and may imply genome-wide or sub-minute data. To maintain technical accuracy, we have removed this term throughout the manuscript and replaced it with more precise language, such as "temporally resolved" This change ensures that our description of the hour-scale ChIP-qPCR intervals and transcriptional analysis aligns with standard nomenclature for kinetic studies.

4. While the temporal uncoupling between transcription and H3K4me3 deposition is clearly demonstrated, the manuscript does not provide mechanistic insight into the functional consequences of this delayed deposition. As such, the study remains descriptive and does not resolve whether H3K4me3 plays a post-transcriptional regulatory role or serves as a passive mark of prior transcription.

Response:

While we acknowledge the foundational studies cited by the reviewer, our work provides a unique temporal map of these dynamics specifically during the rapid induction of human inflammatory genes. The value of this study lies in resolving the precise "lag time" between mRNA synthesis and histone modification, which cannot be captured in steady-state models or lower eukaryotes. We have revised the Discussion to frame our results as a definitive kinetic validation of the "follower" model in a high-demand human transcriptional environment. By establishing this temporal sequence, we provide the essential groundwork for future studies to determine whether this delayed deposition serves a specific post-transcriptional function or acts as a passive marker of activity.

5. The persistence of H3K4me3 following transcriptional inhibition (Figure 5) does not distinguish between delayed deposition and slow turnover of the modification. Therefore, this

experiment does not provide direct evidence supporting a post-transcriptional deposition mechanism.

Response:

We agree with the reviewer that our transcription inhibition at the constitutively active gene *MYC*, the persistence of a pre-existing H3K4me3 after the transcription was blocked, cannot distinguish between delayed loading or due to slow turnover. However, we believe that this experiment provides two critical pieces of evidence: a state of uncoupling of transcription and the H3K4me3 (no transcription but presence of H3K4me3) similar to the uncoupling observed at the inducible genes. Therefore, regardless of the underlying enzymatic cause (loading vs. turnover), our data demonstrate a clear temporal uncoupling between RNA synthesis and H3K4me3 occupancy. This confirms that H3K4me3 levels do not strictly mirror transcriptional kinetics in real-time. In addition, documenting that the mark remains stable for hours after transcriptional shutdown, we provide evidence that the presence of H3K4me3-particularly at genes with low or halted expression-is more accurately interpreted as a signature of prior transcriptional history rather than a marker of current synthesis.

We have moderated our language in the manuscript to ensure we do not claim "post-transcriptional deposition" for the *MYC* model, instead characterizing it as "temporal uncoupling and post-transcriptional retention."

In summary, the manuscript provides a clear temporal analysis of H3K4me3 dynamics at two inducible human loci and supports the view that H3K4me3 follows transcription. However, the conceptual advance relative to prior literature is limited, the conclusions are overgeneralized beyond the data, and key mechanistic claims (e.g., dispensability and post-transcriptional function) are not fully supported by the presented experiments.

We thank the reviewer for acknowledging that the manuscript provides a "clear temporal analysis of H3K4me3 dynamics". While the general relationship between transcription and H3K4me3 is known, we believe the precise kinetic mapping provided here using human inducible genes is a vital empirical contribution to the field. To address the reviewer's concerns, we have revised the manuscript to ensure our conclusions are strictly supported by the temporal data and inducible gene models. Specifically, we have softened our claims regarding "dispensability" and "post-transcriptional function," reframing them as hypotheses

to avoid overgeneralization. These refinements ensure a more focused study that provides a necessary foundation for future mechanistic research.

April 17, 2026

RE: Life Science Alliance Manuscript #LSA-2025-03511-TRR

Dr. Sabarinadh Chilaka
Indian Institute of Chemical Technology
Applied Biology
Tarnaka
Hyderabad, Telangana 500007
India

Dear Dr. Chilaka,

Thank you for submitting your Research Article entitled "Temporal analysis of two inducible human genes reveals post-transcriptional H3K4me3 deposition". It is a pleasure to let you know that your manuscript is now accepted for publication in Life Science Alliance. Congratulations on this interesting work.

Your article will publish open access upon publication under a CC-BY license.

DISTRIBUTION OF MATERIALS:

Again, congratulations on a very nice paper. I hope you found the review process to be constructive and are pleased with how the manuscript was handled editorially. We look forward to future exciting submissions from your lab.

Sincerely,

Sarita Hebbar, PhD
Scientific Editor
Life Science Alliance
<http://www.lsajournal.org>